# Predicting nuclear G-quadruplex RNA-binding proteins with roles in transcription and phase separation

Johanna Luige [1,4], Alexandros Armaos[2,4], Gian Gaetano Tartaglia [2,3] ✉ & Ulf Andersson Vang Ørom [1] ✉

RNA-binding proteins are central for many biological processes and their characterization has demonstrated a broad range of functions as well as a wide spectrum of target structures. RNA G-quadruplexes are important regulatory elements occurring in both coding and non-coding transcripts, yet our knowledge of their structure-based interactions is at present limited. Here, using theoretical predictions and experimental approaches, we show that many chromatin-binding proteins bind to RNA G-quadruplexes, and we classify them based on their RNA G-quadruplex-binding potential. Combining experimental identification of nuclear RNA G-quadruplex-binding proteins with computational approaches, we build a prediction tool that assigns probability score for a nuclear protein to bind RNA G-quadruplexes. We show that predicted G-quadruplex RNA-binding proteins exhibit a high degree of protein disorder and hydrophilicity and suggest involvement in both transcription and phase-separation into membrane-less organelles. Finally, we present the G4-Folded/UNfolded Nuclear Interaction Explorer System (G4-FUNNIES) for estimating RNA G4-binding propensities at http://service.tartaglialab.com/new_submission/G4FUNNIES.

RNA has regulatory and structural roles in all cellular processes that are executed through ribonucleoprotein interactions[1]. The vast interconnection between RNAs and protein factors is reflected in the coordinated cellular responses to external signals or insults. This includes the regulation of transcription, where the interplay of RNA and protein factors controls the assembly of the transcriptional machinery at enhancers and promoters[2]. Accordingly, a growing number of dual specificity DNA–RNA-binding proteins (DRBPs) have been identified[3] and remain a topic of active investigation.

The past decade has provided tremendous insight into RNA-binding proteins by the invention of interactome studies based on poly(A) capture and identification of binding proteins by mass spectrometry[4,5]. Interactome-wide identification has been extended to subcellular compartments[3] and to refined protocols to purify all RNAs independent of poly(A) tails[1]. Compiling the data from experimental studies with human and mouse cell lines as well as tissues suggests that over 6500 mammalian proteins can bind RNA[6]. While some proteins are universally RNA-binding, there are many examples of highly context-specific interactions occurring only in certain stress conditions or within cell types[7–10].

G-quadruplexes (G4) are higher-order nucleic acid structures that form in guanine-rich sequences. The basis for G4 folding is the ability to form hydrogen bonds between two non-adjacent guanines, creating the G-quartet. As a result, the single-stranded nucleic acids can fold into four-stranded G-quadruplexes, formed by the stacking of two or more G-quartets. G-quadruplexes are stabilized by intercalation of monovalent cations between the G-quartets, enhancing the base-stacking interaction.

[1]RNA Biology and Innovation, Institute of Molecular Biology and Genetics, Aarhus University, Aarhus, Denmark. [2]Centre for Human Technologies (CHT), Istituto Italiano di Tecnologia (IIT), Via Enrico Melen, 83, 16152 Genova, Italy. [3]Catalan Institution for Research and Advanced Studies ICREA Passeig Lluis Companys, 23 08010 Barcelona, Spain. [4]These authors contributed equally: Johanna Luige, Alexandros Armaos. ✉e-mail: gian.tartaglia@iit.it; ulf.orom@mbg.au.dk

RNA G4s are dynamic structures, and their function is believed to be regulated by RNA-binding proteins[11–13]. Interaction with RNA G4s can mediate both competition and cooperation between RNA-binding proteins[14]. In vitro, the folding of RNA G4 structures is affected by cations in the buffer, where potassium ($K^+$) stabilizes G4 formation the most, sodium ($Na^+$) provides intermediate stabilization, and lithium ($Li^+$) provides the least stability, which can be exploited in experimental setups to determine G4 function and identify G4-RNA-binding proteins (G4RBP)[15].

Several computational tools exist for identifying potential G-quadruplex forming sequences in human genomic and transcriptomic data[16,17]. These tools primarily focus on the length of the G-tract and the loop in between them to establish the consensus formula $G_{3+}N_{1-7}$ $G_{3+}N_{1-7}$ $G_{3+}N_{1-7}$ $G_{3+}N_{1-7}$, where N represents any nucleotide (A, U/T, G, C) and $G_{3+}$ indicates three or more G nucleotides[18]. However, in vivo, functional G4s may not always conform to this canonical model. For instance, the G-quadruplex in the 5′UTR of *VEGFA* mRNA, validated to be functional in vivo, adopts two alternative conformations of 2-stacked G4s, deviating from consensus sequence[19]. This highlights the diversity and complexity of G-quadruplex structures beyond canonical formulations and underscores the necessity for a nuanced approach to their identification and characterization.

Efforts have been made to identify G4RBPs, with the majority of the data compiled into the QUADRatlas database[13]. This resource offers insights into transcripts with G4-forming RNA sequences and G4RBPs, distributed across various subcellular localizations. However, a significant portion of studies is devoted to interactions involving cytoplasmic G4[20], often overlooking the contribution of DNA binding proteins[21]. An exploration into the nuclear RNA-protein interactome unveiled proteins capable of binding both RNA and DNA, playing a role in the DNA damage response[3]. This discovery suggests a potential for nuclear proteins with dual binding capabilities to engage with G4 structures, coupling RNA and DNA binding possibly through the same domain. Nucleolin (NCL) serves as a prime example of a G4 binding protein able to bind both DNA and RNA, binding to the promoter regions of *VEGFA*[22] and *MYC*[23], as well as interacting with other G4 RNAs[23–25]. This underscores the overarching role of RNA in influencing transcription factor functionality[2].

In this study, we delve deeper into the nuclear RNA-protein interactome. Utilizing a blend of experimental and computational methodologies, we identify G4RBPs and elucidate the specific physicochemical attributes that dictate their affinity for G4 RNA. Our primary tool is the $G_4A_4G_4A_4G_4A_4G_4A_4$ (G4A4) oligonucleotide, previously employed in RNA pulldown experiments[26,27]. Distinct from prior studies that harnessed cytoplasmic or whole-cell extracts[20], we have confined our focus to the nuclear environment, motivated by our objective to profile G4 RNA interactions amongst chromatin-associated proteins specifically. This approach facilitates predictions of proteins that bind to G4 RNA and bolsters our hypothesis regarding the integral role of G4 binding proteins in transcription and phase separation. Our findings accentuate the intricate dance of interactions within the cellular nucleus, offering insights into the multi-dimensional roles of G4RBPs.

## Results

### Chromatin-associated RBPs bind to G4 RNA structures

Given the impact of RNA on transcription factor function[2] and the number of dual DRBPs identified in interactome capture studies[3,6], we wondered whether RBPs associated with chromatin are generally able to directly bind to G4. To identify a suitable G4 forming sequence, we assessed three previously used and characterized oligonucleotides, which represent different classes of G4s – the four-stacked G4A4[26], three-stacked TERRA from telomeric repeat regions[28], and another endogenous G-quadruplex from VEGFA mRNA[19]. Circular dichroism (CD) spectra show a characteristic maximum peak at 265 nm, with a

minimum at 240 nm (Fig. 1a and b, Supplementary Fig. 1a and b), which is evidence for parallel topology, which is most common for RNA G-quadruplexes[29]. Despite the different sequences, all three RNAs form G-quadruplexes that are dependent on $K^+$-stabilization, as the peaks in CD spectra are decreased in $Li^+$ buffer. For G4A4 we recorded CD across a range of temperatures from 20 to 90 °C. Interestingly, we observe similar folding in both ionic conditions for G4A4 at lower temperatures from 20 to 40 °C (Fig. 1a and b), suggesting robust folding and structural stability throughout experimental procedures. Also, G4A4 oligonucleotides have been used to show interaction with the Polycomb repressive complex 2 subunits[26,27]. which supports our choice to use G4A4 for further investigation. We note that sequences such as G4A4 and G3A2[20], which form G-quadruplexes with four and three perfectly stacked quartets, are present in a minor proportion (2.5% and 25%, respectively) within the QUADRatlas database[13]. Yet, G4A4 demonstrates greater stability compared to VEGFA[19], and TERRA[28] G4s, which is instrumental in determining its precise interactome.

We utilized the *cat*RAPID approach to assess the affinity of 283 chromatin-related RBPs, identified in K-562 nuclei, towards G4 RNA structures[3,30]. Our analysis differentiated the binding preferences of these proteins between folded and unfolded G4 RNA states, facilitated by $K^+$ and $Li^+$ ions, respectively (see the "Methods" section). Remarkably, 182 proteins exhibited a preference for the structured G4 RNA, underscoring the extensive inclination of chromatin-associated RBPs towards these conformations (Fig. 1c, Supplementary Data 1).

### Experimental identification of G4RBPs

Next, we proceeded to experimentally assess the binding preference of RBP through an in vitro approach (see the "Methods" section). In our experiments, we exploited the G4A4 oligo as used for computational modeling and coupled it to biotin to allow for the purification of bound proteins from K-562 nuclear extract. We performed the experiment in the presence of $K^+$ or $Li^+$ cations to affect the stability of G-quadruplex structures in the G4A4 oligoribonucleotide[29]. To set up the conditions and demonstrate the sensitivity of our approach to purify G4RBPs, we used the well-characterized G4 RNA binder NCL[23–25]. The protocol for protein purification is shown as a diagram in Supplementary Fig. 1c. We can detect NCL binding to G4A4 by western blot (Fig. 2a), which is dependent on $K^+$ ions in the assay buffer. Across four independent experiments, significantly higher levels of NCL are observed in $K^+$ compared to $Li^+$ conditions (Fig. 2b), which is in agreement with the known impact of cations on G4 RNA structure stability[15], and shows that the method can be used for large-scale purification of G4RBPs. We then subjected samples to LC–MS/MS mass spectrometry in triplicate and identified proteins that were significantly enriched or depleted in $K^+$ buffer compared to $Li^+$ buffer (Fig. 2c). By mass spectrometry, we detected a total of 1204 proteins, of which 151 and 83 are enriched in the purification using $K^+$ and $Li^+$ cations in the binding buffer (Supplementary Data 2; see the "Methods" section).

### G4RBPs are generally known RBPs

Of the 151 proteins with increased binding upon G4 stabilization, 83 (55 percent) have a previous annotation as G4RBP according to QUADRatlas[13], while for the proteins with decreased binding upon G4 stabilization, 26 (31 percent) are classified as G4RBP by QUADRatlas (Fig. 3a, Supplementary Data 3), demonstrating that our assay enriches for bona fide G4RBPs (p-value < 0.001; Fisher-exact test). Of the 68 proteins not previously shown to associate with G4 RNA, 19 have Gene Ontology (GO) term DNA binding (p-value < 0.0003; Student's t-test), including CTCF and TOP1, pointing towards G4-binding as a bridge between DNA and RNA binding properties for chromatin-associated proteins. At the same time, this comparison shows that several aspects of proteins and G4-forming RNA sequences are likely to impact their binding. When GO for the entire set of identified proteins, we see that

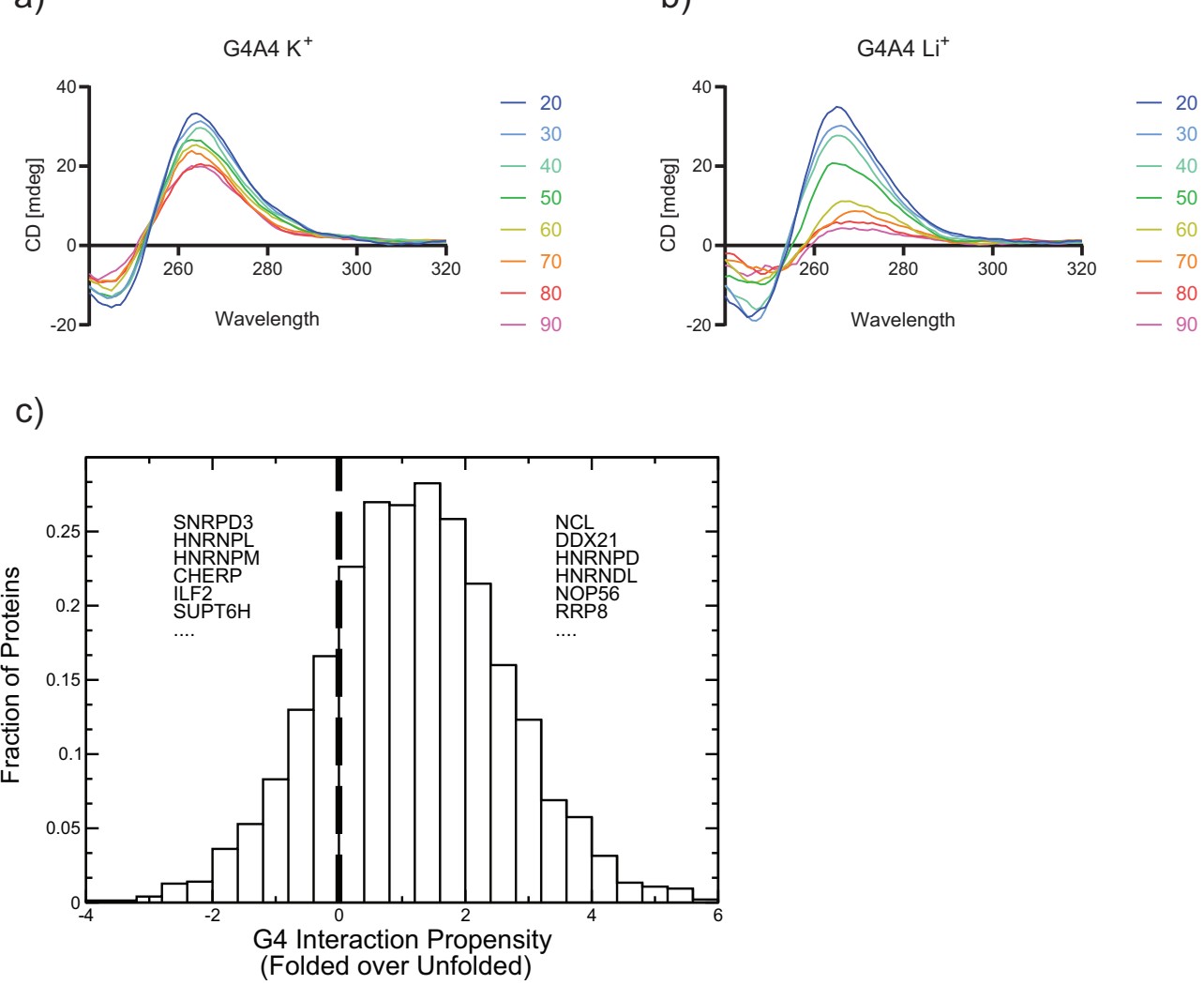

**Fig. 1 | Chromatin RBPs are predicted to bind to the G quadruplex structure. a** Circular dichroism spectrum in a temperature gradient for G4A4 RNA in K$^+$ buffer. Source data are provided as a Source Data file. **b** Same as in (**a**) using Li$^+$ buffer. Source data are provided as a Source Data file. **c** *cat*RAPID predictions of RBP interactions with folded (structured) and unfolded (linear) G4A4 indicate that most chromatin-related proteins have a binding preference for folded G4A4. Examples of proteins with a preference for folded and unfolded G4A4 are reported. Source data are provided as a Source Data file.

the overall identified proteins in the mass spectrometry analysis, regardless of response to K$^+$ and Li$^+$, show 48.3 percent with the GO term RNA binding (566 of 1172 with annotated GO term) (Fig. 3b), supporting the quality of our in vitro pull-down assay. For proteins that are enriched in K$^+$ buffer, this percentage increases to 79.9 (119 of 149 with annotated GO term), and for proteins depleted in K$^+$ buffer compared to Li$^+$ buffer this percentage is decreased to 22.9 (Fig. 3b). These data show that the proteins with increased binding in K$^+$ buffer are bona fide RNA binding proteins where most have been annotated with the GO term RNA binding, supporting G4 RNA binding as a central property for RBPs. In contrast, proteins that are depleted in K$^+$ buffer are non-canonical RBPs. Overlap with the nuclear interactome from K-562 cells[3] shows that 48 of the proteins increasing binding in the K$^+$ buffer overlaps with the previously annotated K-562 nuclear interactome, whereas only a single protein depleted in K$^+$ buffer does, showing that enrichment of binding in response to K$^+$ to stabilized G4 RNA sequences supports an annotation as true RBPs (Fig. 3d). In total, 229 of the identified proteins overlap with the 343 proteins (67.8 percent) in the K-562 nuclear interactome. 766 proteins neither overlap with the K-562 nuclear interactome nor change their binding in

response to K$^+$ and Li$^+$, and amongst these, 254 [out of 761 (33 percent) with annotated GO term] have the GO term RNA binding, suggesting that these might predominantly consist of background due to the in vitro nature of the assay.

Of the 31 proteins (30 with annotated GO term) not annotated as RNA-binding that are enriched in K$^+$ buffer, 12 have the GO term transcription, and 12 have the GO term DNA binding with a substantial overlap between the two groups (Fig. 3c). In the K-562 nuclear interactome not overlapping with the proteins identified in this study there is no significant enrichment for DNA-binding proteins or transcription factors, suggesting that G4RBPs does have an important role for connecting RNA- and DNA-binding proteins (Fig. 3d).

Upon analyzing proteins binding to G4 RNA in the presence of K$^+$ contrasted to those in the presence of Li$^+$, we observed a significant enrichment of specific PFAM domains. Prominently, the DEAD/DEAH box proteins and Helicase C emerged. This group comprises proteins such as DDX21, DDX42, DDX24, DDX18, and DDX56. Additionally, MTREX is associated with the DEAD category, while CHD7 is linked to Helicase. In the presence of K$^+$, these motifs show significant enrichment, with adjusted *p*-values falling below 0.10 (hypergeometric test;

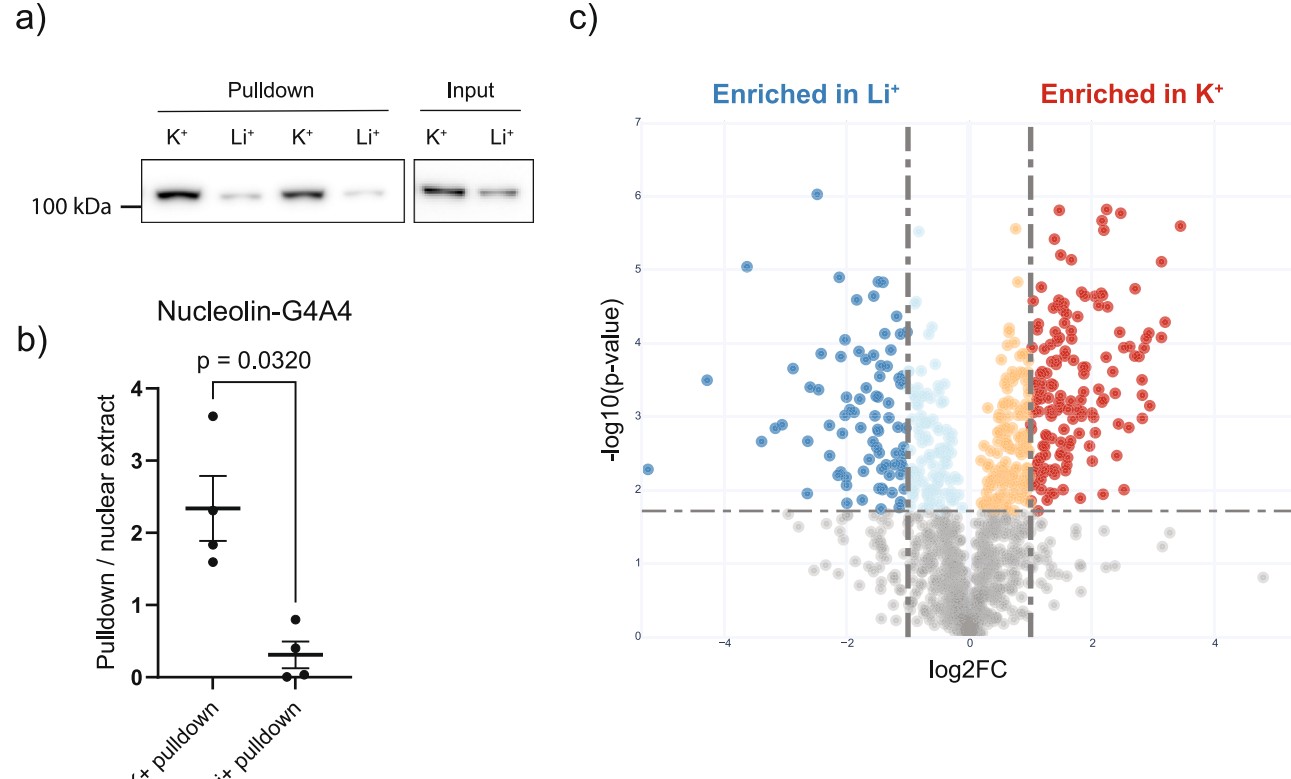

**Fig. 2 | Experimental identification of G4-RBPs. a** Western blot for NCL after pulldown of NCL from nuclear extract with G4A4 RNA, displaying *n* = 2 representative experiments. **b** Quantification of Western blot from four pulldown experiments represented as a ratio of pulldown to input protein levels, showing NCL binding is dependent on folded G4 (K⁺). Data are presented as mean values ± SEM of *n* = 4 independent experiments, with statistical significance level determined by a two-tailed paired *t*-test (*p* = 0.032). **c** Volcano plot for significantly enriched proteins in K⁺ and Li⁺ pulldown mass spectrometry experiment (Supplementary Data 2). Statistical significance was estimated by unpaired *t*-test, with *p*-value < 0.05 determined as significant. Source data are provided as a Source Data file.

see the "Methods" section). Importantly, helicases are recognized for their affinity to G4. Indeed, DDX21 has been documented to bind and unwind RNA guanine quadruplexes[31]. In contrast, no specific PFAM domains were discerned when contrasting the protein groups in Li⁺ and K⁺ environments.

## Computational characterization of G4RBPs

Having established that proteins enriched in K⁺ buffer are bona fide RBPs enriched in conditions where the folding of G4A4 is stabilized into a G4 structure, we set out to explore the properties of these protein groups.

First, we evaluated *cat*RAPID's[32] performance in predicting interactions with experimentally identified G4RBPs. Our analysis included proteins found to interact with folded (K⁺ group) or unfolded (Li⁺ group) G4A4. The plot of cumulative K⁺ over Li⁺ group protein enrichment against the differential score of folded over unfolded G4A4 indicates that as the propensity for folded G4A4 interaction increases, there is a corresponding rise in K⁺ group proteins, aligning with experimental results and validating catRAPID's predictive accuracy (Fig. 4a, Supplementary Data 4; see the "Methods" section).

We next sought a deeper comprehension of G4RBP characteristics using the *clever*MACHINE, an algorithm distinguishing between two unique protein datasets by evaluating the intrinsic physicochemical properties in their sequences (see the "Methods" section)[33]. The focus of the analysis is proteins interacting with G4A4 in a K⁺ environment, compared against those in Li⁺ settings, aiming to extrapolate their differences. The application of *clever*MACHINE shows a 96% confidence level differentiation between the datasets (Supplementary

Data 4). We observe a notable enrichment of non-classical RBPs (Area under the ROC Curve, AUC, 0.88, Fig. 4b) in proteins present in K⁺ conditions. In contrast, there is a marked depletion in burial (AUC 0.81; Supplementary Fig. 2a) and hydrophobicity (AUC 0.77; Fig. 4c). As the *clever*MACHINE indicates a change in structural disorder (B-value propensity score, AUC 0.71; Supplementary Fig. 2a), we used Alpha-Fold to provide a refined analysis of structural content within these protein sets, confirming the pronounced disorder enrichment in K⁺ associated proteins (Fig. 4d; Supplementary Data 5).

Using the *clever*MACHINE, we developed the G4-Folded/UNfolded Nuclear Interaction Explorer System (G4-FUNNIES) and have made this resource publicly available. The users can access G4-FUNNIES to evaluate RNA G4-binding propensities of proteins via the following link: http://service.tartaglialab.com/new_submission/G4FUNNIES.

## Phase separation propensity of G4RBPs

Disordered protein domains are known for their multifaceted roles in the formation of biomolecular condensates, in which protein and nucleic acids within a solution can dynamically undergo demixing, resulting in separation into distinct phases with different molecular compositions[34]. We investigated the phase-separation ability of proteins using *cat*GRANULE[35] and found that proteins binding to G4A4 in the presence of K⁺ have, indeed, higher phase separation propensity (Fig. 4e; see the "Methods" section, Supplementary Data 6). We complemented our predictions with an analysis of protein occurrence in phase-separated organelles[36] (Supplementary Data 7). In agreement with our predictions, proteins binding to G4A4 in the presence of K⁺ show enrichment across the condensation states of nucleolar proteins

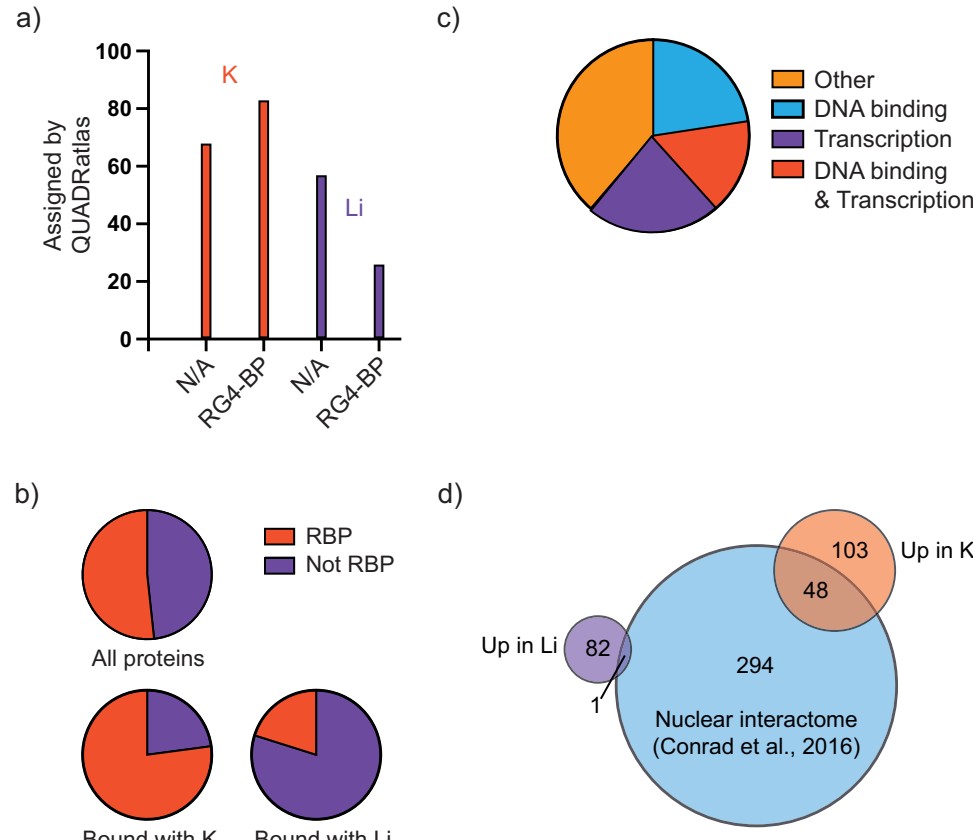

**Fig. 3 | Characterization of proteins identified in G4A4 pull-down. a** Comparison to QUADRatlas of the proteins identified with pull-down and mass spectrometry. Graph shows how many proteins for each of the K[+] and Li[+] pull-down conditions have been assigned as G4RBP (RG4-BP) or have not been shown to bind to G4 RNA sequences (N/A). **b** Analysis of GO term RNA-binding protein (RBP) for identified proteins in the three groups all proteins (1204), Bound with K[+] (151), and Bound with Li[+] (83). **c** GO-analysis of proteins bound in K not annotated as RBP. Source data are provided as a Source Data file. **d** Overlap with NIC. A Venn diagram showing the overlap between RBPs identified by nuclear interactome capture and by binding to the G4A4 sequence in a stabilized (up in K[+]) or destabilized (up in Li[+]) state. Source data are provided as a Source Data file.

(Fig. 4f)[36], thus confirming *cat*GRANULE calculations. Indeed, recent studies suggest G4 can drive phase separation[37,38], particularly in nucleolus[39,40], by allowing multivalent interactions through potential stacking of multiple folded G4s[41], as well as providing spatial recognition surfaces for protein partners[42].

Given that phase separation is regulated by post-translational modifications (PTMs)[43,44], we delved into analysis to discern if distinct PTMs characterize proteins that interact with G4 in the presence of K[+] or Li[+]. Specifically, we used the ELM database (http://elm.eu.org/) to identify experimentally validated or predicted PTMs[45]. In the group of proteins interacting with G4 in the presence of K[+], there was an enrichment of sumoylation, specifically associated with MOD_SUMO_rev_2 or ELME000393 (Supplementary Fig. 3). Given that sumoylation plays a crucial role in numerous nuclear functions and facilitates significant subnuclear relocations of the modified proteins, its influence on phase separation is relevant[46]. Conversely, the group of proteins interacting with G4 in the presence of Li[+] exhibited an enhanced presence of sites for Cdc14 phosphatase dephosphorylation, denoted by MOD_CDC14_SPxK_1 or ELME000529 (Supplementary Fig. 3). Since phase separation is modulated by phosphorylation events[47], this enrichment is particularly noteworthy. Moreover, an increased inclination for phosphorylation, characterized by MOD_ProDKin_1 or ELME000159, was distinctive of the K[+] group. Though both sumoylation and dephosphorylation were enriched at experimental and predicted levels, a significant conundrum emerges in discerning the actual PTMs that are operational within a cellular milieu, specifically modulating individual proteins.

## Computational identification of G4RBPs

We extended the study beyond the G4A4 interactome using *clever*MACHINE[33] to analyze another G4 RNA, G3A2, previously characterized in terms of its cytoplasmic interactome (see the "Methods" section; Supplementary Fig. 2b)[20]. Also, in this case, the *clever*MACHINE effectively distinguishes G4RBPs from guanine-binding RBPs (see the "Methods" section)[20]. Intriguingly, we found a convergence in the physicochemical property patterns between the G3A2 predictor and G4-FUNNIES, revealing a universal signature characterizing G4 binding proteins irrespective of subcellular localization distinctions (Supplementary Fig. 2b). Yet, due to the varying properties of proteins across different environments, our algorithm developed using G3A2 is less precise in discriminating between interactions with folded and unfolded G4 structures in G4A4 (see the "Methods" section).

In the analysis of G4A4 and G3A2, we distinguish between the folded and unfolded G4 groups. To enhance the precision of our G4-FUNNIES tool, we incorporated a filter that initially determines whether a particular protein set engages with G4 RNA (i.e., G4 binding ability). The method we built utilizes the protein set that consistently ranks lowest in our mass spectrometry data (Supplementary Data 2; see the "Methods" section). In Fig. 5a, we show that when eCLIP-identified RBP target occurrences of G4 RNA are processed using the G4 RNA propensity score of *pqsfinder*[48], a clear distinction emerges between G4 binders and non-binders according to the *clever*MACHINE classification (Supplementary Data 8). This division is notably pronounced with elevated scores from the G4 RNA prediction tool,

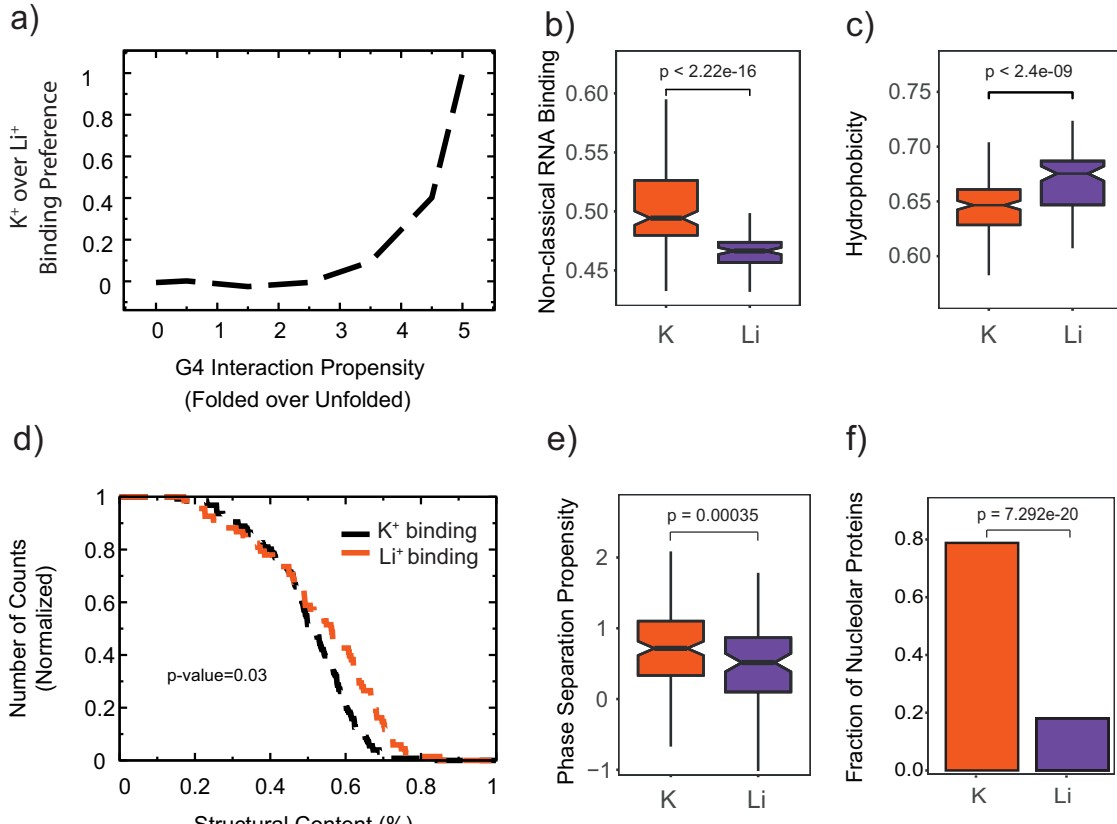

**Fig. 4 | Physico-chemical properties of proteins interacting with G4. a** As the interaction propensity for folded G4A4 increases, we observe a corresponding increase in the cumulative enrichment of $K^+$ group proteins over $Li^+$ group proteins, underscoring the agreement of our model with the experimental results. Source data are provided as a Source Data file. **b** A significant number of proteins interacting with G4A4 in the presence of $K^+$ ions are predicted to be non-classical RBPs[7], and **c** depleted in hydrophobicity (see Supplementary Fig. 2)[84]. **d** Structural content in $K^+$ and $Li^+$ protein groups as determined by AlphaFold (see Supplementary Data 5). **e** Phase separation propensity as calculated with *cat*GRANULE (see Supplementary Data 6). **f** Condensation state of proteins interacting with G4A4

preferentially with $K^+$ or $Li^+$ (see Supplementary Data 7). In **b**, **c**, and **e**, the boxes depict the interquartile range (IQR), the central line denotes the median, and the notches represent the 95% confidence interval of the median. The whiskers extend from the box by adding 1.5 times the IQR to the 75th percentile (upper limit of the box) and by subtracting 1.5 times the IQR from the 25th percentile (lower limit of the box). The sizes of the $K^+$ and $Li^+$ protein groups are reported in Fig. 3, and differences between sets are evaluated using a two-tailed *t* test. For **f** the difference is assessed using Fisher's exact test. $K^+$ group proteins and $Li^+$ group proteins are indicated in red and purple, respectively.

underscoring the ability of our *clever*MACHINE in segregating the two groups. Figure 5b further emphasizes this distinction, revealing that as the confidence score of the *clever*MACHINE escalates, a corresponding increase in the detection of G4 RNA is observed across the two protein categories (those that bind to G4 RNA versus those that do not). Such a pattern attests to our model's capability to precisely pinpoint significant G4 RNA signals, as corroborated by eCLIP experimental data. We note that as the G4 confidence intensifies, there is an augmented inclination towards folded G4 (Supplementary Data 8).

**Experimental validations of candidate G4RBPs**

To better investigate the nuclear repertoire of RBPs, we carried out further analysis with G4-FUNNIES. Specifically, we computed the G4-interaction propensity to all proteins with the GO term 'chromatin' (0000785, 1186 proteins, of which 561 are predicted G4-binding; Supplementary Data 9). We used G4-FUNNIES to calculate interaction propensities for chromatin proteins and classified them as $Li^+$ and $K^+$, with a lower cut-off for scoring as G4RBP set at 50 (Supplementary Data 9; see the "Methods" section), assigning G4RBP properties (i.e., G4 binding ability and G4 structural preference; Fig. 6a). In line with what is reported in QUADRatlas[13] and our mass-spec experiments, histones (e.g., H3C14), helicases (e.g., DDX21), and exosome components (e.g., EXOSC3) exhibit a high G4 binding affinity as well as a pronounced preference for structured G4. As in the previous analysis,

*cat*GRANULE predictions indicate enrichment in phase separation propensities of proteins predicted to bind G4 RNA in the presence of $K^+$ (*p*-value < 0.000001; Kolmogorov–Smirnov test; Supplementary Data 10).

To validate whether the G4-FUNNIES candidates we assign as G4RBP bind preferentially to G4 RNA, we pulled down RUVBL2 from nuclear extracts using G4-forming G3A2 and its unstructured counterpart G3 MUT RNA (Fig. 6b). We see a significantly higher enrichment of RUVBL2 when using the structured G4-forming G3A2 than with the G3 MUT that does not form structured G4 (Fig. 6c), supporting that RUVBL2 binds to various G4-forming sequences and underlines the contribution of proper structure of the folded RNA for efficient binding to G4RBPs. Next, we assessed the binding of RUVBL2 to G4 RNA sequences within the cell by native RNA immunoprecipitation followed by RT-qPCR analysis of protein-bound RNAs (Fig. 6d; see the "Methods" section). We use *VEGFA*, *MYC*, *BCL-2*, and *NRAS* as references for endogenous G4 RNA, as these are some of the most thoroughly validated and prominent cellular mRNAs harboring 5′UTR G-quadruplexes[18,19,49,50]. The negative control RNAs *RN7SK* and *GAS5* are chosen for their apparent lack of annotated and predicted G4s according to the QUADRatlas and being highly abundant transcripts. We can see significant enrichment to RUVBL2 for all four G4 mRNAs, while no interaction with the control RNAs was detected over background (Fig. 6d). Taken together, these results show the predictive

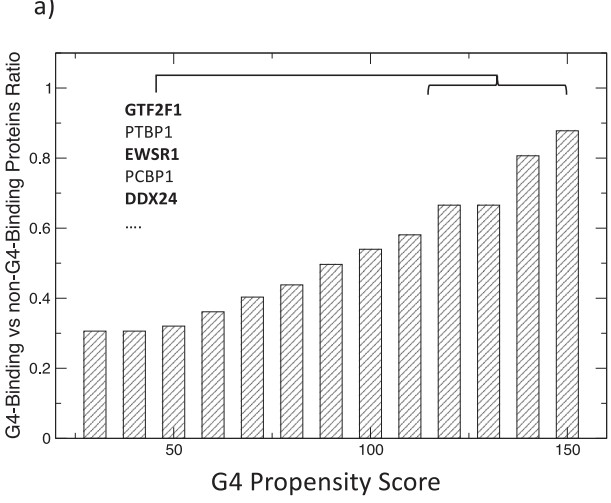

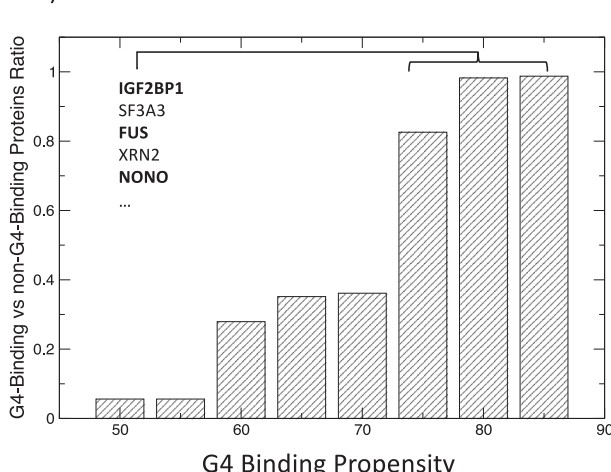

**Fig. 5 | Prediction of G4 binders vs. non-binders based on eCLIP-identified RBP targets. a** Enhanced differentiation of G4 binders from non-binders is associated with rising G4 RNA propensity scores. **b** The differentiation intensifies with increased *clever*MACHINE confidence levels. For both analyses, 'the G4-binding to non-G4-binding protein ratio is determined' by the count of G4s associated with each category. Proteins predicted to bind folded G4 are written in bold. See also Supplementary Data 8.

potential of G4-FUNNIES and validate RUVBL2 binding preference for G4 RNA.

## Discussion

RBPs have been investigated at a large scale in the last decade following the invention of poly(A) mRNA interactome capture[4,5]. Approaches to identify proteins binding to all RNA have subsequently been developed[1]. Common to these existing approaches is that they do not consider RNA structural elements, such as G-quadruplexes, that are context-dependent and responsive to *e.g.* cations[15] and stress[51]. In this study, we show by experimental large-scale identification of G4RBPs that several proteins previously not detected to bind RNA in K-562 nuclei[3] are RNA binding and use computational tools to expand to the hypothetical complete G4RBP proteome. DRBPs have been identified by serial interactome capture of the cell nucleus to be a poly(A) mRNA enriched class of proteins[3], suggesting an intimate relationship between DNA- and RNA-binding properties for transcription factors and DNA damage proteins, where RNA could facilitate DNA binding and modulate enzymatic activity, as recently suggested to be a general feature for transcription factors[2]. G4 forming sequences are present, particularly at enhancers, promoters, and within 5'UTR encoding sequences, and are thus well-positioned in the genome and transcriptome to link DNA- and RNA-binding properties for specific groups of proteins.

We predicted and validated interactions with several proteins, some not previously associated with G4 binding. Certain proteins that we link to unfolded G4 have been reported in other studies to interact with G4. We attribute these inconsistencies to three primary factors. First, our study focuses on the nucleus, suggesting that the cellular environment may influence G4 binding affinities. Proteins might exhibit different binding behaviors depending on their cellular location. This specificity adds a layer of complexity to the understanding of G4-protein interactions. Second, the diversity in G4-forming RNA sequences used across studies contributes to these discrepancies. We detect a level of overlap with datasets combined into the QUADRatlas database while also uncovering proteins previously not associated with G4-binding. Additionally, we and others have found differential binding to endogenous G4 mRNAs[52], implying that the selection of G4 RNA is an important determinant of the identified proteome. Third, experimental variations, including different protocols and mass spectrometry techniques, introduce another level of complexity.

These methodological differences can lead to variations in identified G4-binding proteins. Yet, when comparing our findings with the G3A2 cytoplasmic interactome[20], we noticed substantial similarities. Our development of a predictor, informed by this cytoplasmic data, revealed physicochemical traits consistent with our G4A4 predictor. Traits such as hydrophilicity and disorder, indicative of phase separation[35], were prevalent in G4 binding proteins. These consistent patterns emphasize the shared characteristics of G4 binding proteins, highlighting the need for different approaches to unravel these complex interactions. It is crucial to mention that protein binding to G4 typically occurs with varying affinities that are within the micromolar range[29,51]. This affinity range potentially underscores the establishment of transient, weak interactions instrumental in phase separation processes[53,54]. In addition, we must consider the potential influence of post-transcriptional modifications[44]. Such modifications can impact not only the cellular localization but also the properties required to interact with G4s, necessitating further exploration to fully understand their role in protein function[43].

In addition, we propose that G4RBPs can accumulate in membrane-less organelles as the nucleolus, where binding of G4RBPs to mRNAs could mediate an efficient regulation of translation and localization of regulatory intracellular bodies. Indeed, through high-throughput dimethylsulfate probing, it has been shown that the G4 structure forms upon stress[55], and several reports indicate that G4 sequences induce phase separation[14,37]. We note that the proteomes of the nucleolus are characterized by proteins with a higher degree of intrinsically disordered regions (IDRs), as well as well-known RNA-binding domains, such as RRG/RG motif[56], RRM, and DEAD domains[56,57]. Moreover, studies of DDX ATP-dependent helicases in multiple species show their ability to induce phase separation through low complexity protein domains, determined by the ATP hydrolysis state and the resulting interactions with their RNA substrates[58]. Most importantly, this property of DDX helicases allows for the turnover of the membrane-less compartments and can facilitate RNA partitioning between different granules, both in the cytoplasm and nuclear environment.

Several interesting examples of G4RBPs result from our analysis, including the enhancer-binding protein CTCF that has been shown to bind G4 DNA[59] and to require RNA for recruitment to chromatin[60]. We also found YY1, which is central to enhancer function and has been reported to bind G4 DNA[61]. From these data it is tempting to speculate

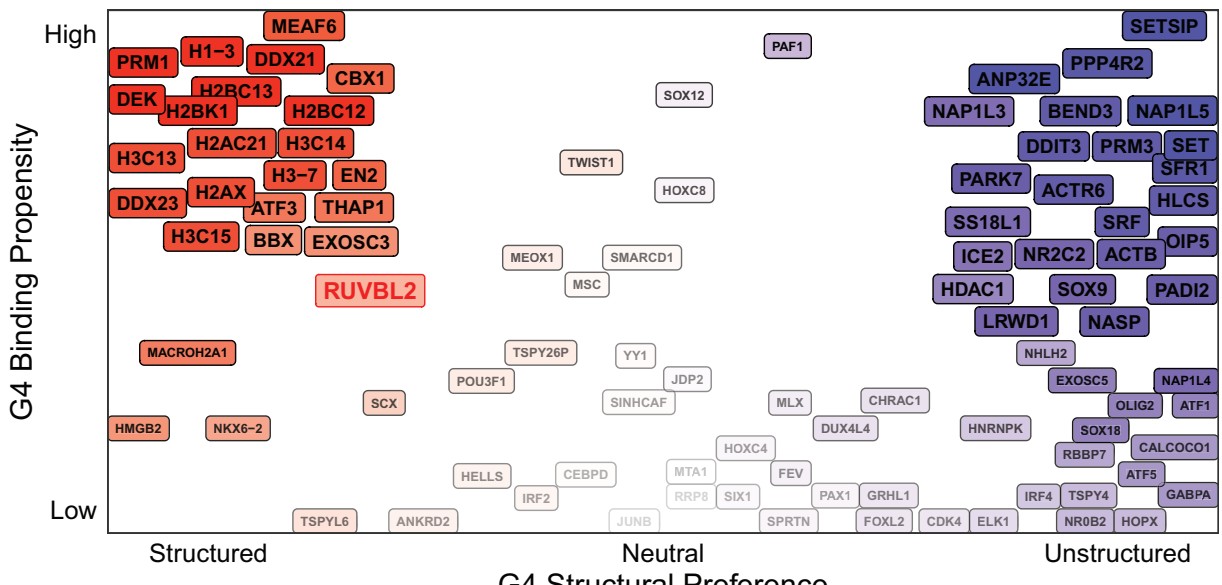

a)

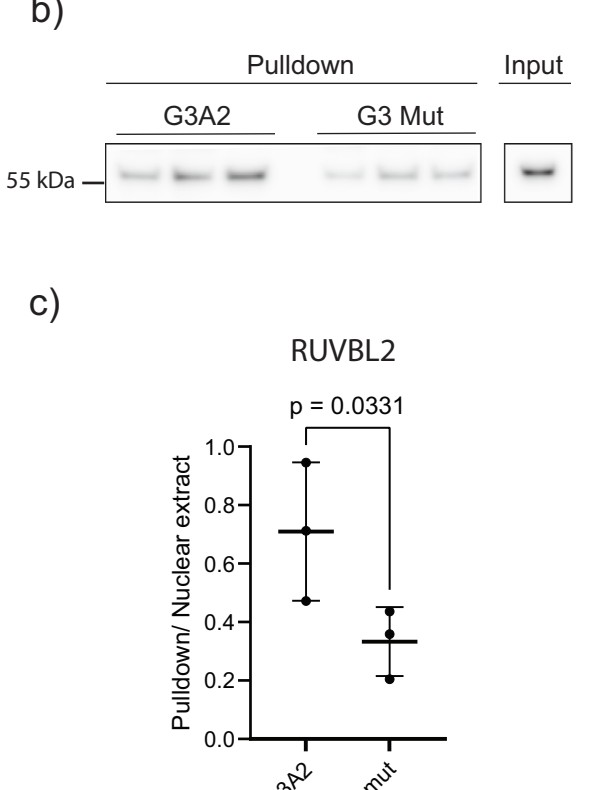

b)

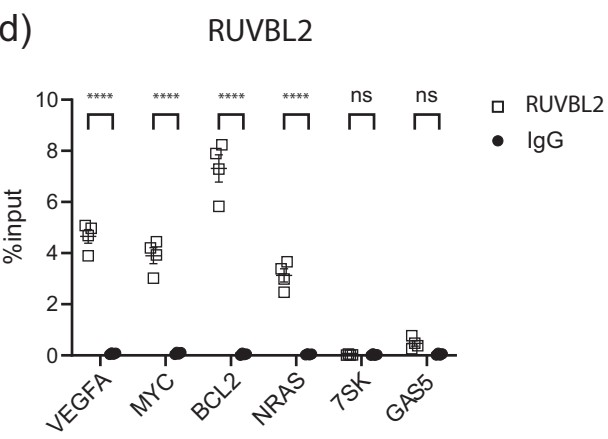

d)

that G4 at transcribed enhancers could facilitate transcription factor recruitment with impact on both enhancer function and chromatin landscape. At another cellular level, we predict TOP1 to be a G4RBP. TOP1 has recently been shown to bind G4 at DNA[62] and facilitate transcription of the *MYC* gene, which encodes a well-characterized G4-containing mRNA. Finally, we suggest 19 proteins involved in DNA damage to be G4RBPs, including CHEK1. CHEK1 is a cell-cycle checkpoint kinase that has also been shown to be involved in DNA double-strand break repair[63]. The recognition of transcribed G4 RNA at DNA

damage sites could be one of the underlying aspects of RNA requirement for DNA damage repair that has been reported[64]. Moreover, the formation of G-quadruplexes has been linked to R-loops, with both secondary structures marking damage-sensitive genomic regions[65,66], as well as being recognized by DEAD-box helicases[64], which were also significantly enriched in our data.

In conclusion, we identify a set of bona fide RBPs that recognize G4 RNA and use the biophysical properties of these proteins to model which chromatin-binding proteins have G4 RNA-binding propensity.

**Fig. 6 | Analysis of chromatin protein interactions with G4 RNA. a** We constructed a global ranking for proteins predicted to associate with both structured and unstructured G4 (x-axis) and plotted it against the G4 binding ability (y-axis; see Supplementary Data 9). Color transparency correlates with the scores represented on the axes: lack of transparency signifies high scores, while high transparency denotes low scores. Red and purple signify proteins predicted to have structured and unstructured G4 binding preferences, respectively. **b** Western blot for RUVBL2 after pulldown of RUVBL2 from nuclear extract with G3A2 and its unstructured counterpart (G3 MUT) RNA oligonucleotides in $n = 3$ individual pulldown experiments. Source data are provided as a Source Data file. **c** Quantification of Western blot from three independent experiments showing RUVBL2 binding preferentially to G4-forming RNA G3A2, represented as the ratio of pulldown to input protein

levels, corresponding to an average 1.1 percent of input for WT and 0.5 percent of input for MUT. Individual datapoints of $n = 3$ experiments are shown as mean values ± SEM, statistical significance ($p < 0.05$) determined by two-tailed paired $t$-test ($p = 0.0331$). Source data are provided as a Source Data file. **d** Native RIP-qPCR experiments for RUVBL2, validating G4RBP interaction with *VEGFA*, *MYC*, *BCL-2*, and *NRAS* mRNAs, as well as showing no binding to *7SK* and *GAS5* RNAs as control, represented as %input for RUVBL2 and IgG control. Individual RNA-immunoprecipitation experiments $n = 4$ are shown with mean values ± SEM, statistical significance levels estimated using two-way ANOVA, with Šídák's multiple comparisons test (ns = not significant; $*p < 0.05$; $**p < 0.01$; $****p < 0.0001$). Source data are provided as a Source Data file.

We provide an overview of DRBPs that could connect DNA to RNA in biological processes as enhancer function, transcription, and DNA damage repair; show evidence of G4 RNA-mediated localization of proteins to nucleolus; and present the biophysical properties and protein motifs important for recognition of RNA G4 structured sequences, that can be accessed as a tool on the webserver http://service.tartaglialab.com/new_submission/G4FUNNIES.

## Methods

### G-quadruplex RNAs
G4A4 AAAAAAGGGGAAAAGGGGAAAAGGGGAAAAGGGGAAAAAA
TERRA UUAGGGUUAGGGUUAGGGUUAGGG
VEGFA GGAGGAGGGGGAGGAGGAAGA
G3A2 UCUGGGAAGGGAAGGGAAGGGAUC
G3 MUT UCUGCGAAGCGAAGCGAAGCGAUC

### Tissue culture
K562 human myelogenous leukemia cell line (ATCC CCL-243) was obtained from ATCC. Cells were cultured in RPMI-1640 Glutamax (61870-036, Thermo Fisher Scientific) medium supplemented with 10% FBS (Gibco), 100 U/ml penicillin and 100 µg/ml streptomycin. K562 cells were maintained in $2 \times 10^5$ to $1 \times 10^6$ viable cells/ml.

### Circular dichroism spectroscopy
Circular dichroism for G4A4 RNA was carried out with a Jasco J-810 spectropolarimeter. CD analysis of 2.5 µM RNA was carried out in the buffer used for G4-pulldown, containing 25 mM HEPES pH 7.0, 150 mM KCl or LiCl, and 5% glycerol. Spectral signatures were recorded over 220–320 nm wavelength. For VEGFA and TERRA, G4 RNA CD spectra were obtained at 25 °C. To show the full extent of structural stability, the CD spectrum of G4A4 oligonucleotide was recorded in a range of temperatures from 20 to 90 °C

### G-quadruplex pull-down
K562 cells were harvested and washed in PBS before lysis. Initial cytoplasmic lysis of K562 cells was carried out using an isotonic lysis buffer 0.1% Igepal CA-630, 150 mM NaCl, 50 mM Tris pH 8.0, supplemented with protease inhibitor cocktail (P8340, Sigma-Aldrich) and 1 mM PMSF, incubating on ice for 10 min. Nuclei were collected by centrifugation for 3 min at 300 g. After removing the cytoplasmic fraction, nuclei were suspended in K$^+$ or Li$^+$-containing Buffer D (10 mM HEPES, 100 mM KCl/LiCl, 5% glycerol, 1 mM PMSF, 1% protease inhibitor cocktail), incubated on ice for 10 min. 3 times 30 s ON/OFF sonication was carried out at high intensity (Covaris S2 Focus Ultra-sonicator) to disrupt the chromatin and release nuclear proteins. Lysates were centrifuged for 30 min at 4 °C at 17 000×g.

5′ biotinylated oligonucleotides were received from Integrated DNA Technologies, or alternatively 3′ biotinylation of G-quadruplex forming RNA oligonucleotides was carried out in 30 µl reaction, with 100 pmol RNA, 2 nmol biotinylated cytidine (19519016, Thermo Fisher Scientific), 2U T4 RNA ligase (EL0021, Thermo Fisher Scientific) in 1X ligase buffer, 20 U SUPERaseIN RNase inhibitor and 15% final

concentration of PEG-8000. Ligation was incubated at 16 °C for 16 h. Biotinylation reaction was cleaned with chloroform-isoamyl alcohol (24:1) extraction and ethanol precipitation. Pulldowns were carried out in either K$^+$ or Li$^+$ containing buffers (10 mM HEPES (pH 7.9), 150 mM KCl or LiCl, 5% glycerol, 0.2 U/µl SUPERaseIN), maintaining the buffer conditions throughout the experiment. Biotinylated RNAs were folded in K$^+$/Li$^+$ buffers by heating at 95 °C for 5 min, followed by incubation at room temperature for 1 h.

Neutravidin-coated magnetic beads (Cytiva 78152104011150) were washed in K$^+$/Li$^+$ buffers 3 times 5 min with rotation before the addition of the biotinylated G4-RNA.

Each pulldown was carried out with 150 pmol biotinylated RNA, 50 µl beads, and 300 µg of K562 nuclear extract, incubating overnight on rotation at 4 °C. After incubation, beads were washed with K$^+$/Li$^+$ pulldown buffer 3 times 10 min on rotation at 4 °C.

### AP-MS
Washed beads were incubated for 30 min with elution buffer 1 (2 M Urea, 50 mM Tris–HCl pH 7.5, 2 mM DTT, 20 µg/ml trypsin) followed by a second elution for 5 min with elution buffer 2 (2 M Urea, 50 mM Tris–HCl pH 7.5, 10 mM chloroacetamide). Both eluates were combined and further incubated at room temperature overnight. Tryptic peptide mixtures were acidified to 1% TFA and loaded on Evotips (Evosep). Peptides were separated on 15 cm, 150 µM ID columns packed with C18 beads (1.9 µm) (Pepsep) on an Evosep ONE HPLC applying the '30 samples per day' method and injected via a Captive-Spray source and 10 µm emitter into a timsTOF pro mass spectrometer (Bruker) operated in PASEF mode[67].

### Enrichment analysis of Pfam domains
We retrieved Pfam annotations from the Enrichr database[68] in order to detect domains enriched in the K+ against the Li+ set. We performed Over-Representation Analysis (ORA) using the R package clusterProfiler (v4.2.2).

### Data analysis
Raw mass spectrometry data were analyzed with MaxQuant (v1.6.15.0). Peak lists were searched against the human Uniprot FASTA database combined with 262 common contaminants by the integrated Andromeda search engine. False discovery rate was set to 1% for both peptides (minimum length of 7 amino acids) and proteins. "Match between runs" (MBR) was enabled with a Match time window of 0.7 and a Match ion mobility window of 0.05 min. Relative protein amounts were determined by the MaxLFQ algorithm with a minimum ratio count of two. All statistical analysis of LFQ-derived protein expression data was performed using the automated analysis pipeline of the Clinical Knowledge Graph[69]. Protein entries referring to potential contaminants, proteins identified by matches to the decoy reverse database, and proteins identified only by modified sites were removed. LFQ intensity values were normalized by log2 transformation and proteins with less than 70% of valid values in at least one group were filtered out. The remaining missing values were imputed using the MinProb

approach (random draws from a Gaussian distribution; width = 0.2 and downshift = 1.8)[70]. Differentially enriched proteins in each group comparison were identified by SAMR multiclass test with permutation-based FDR correction for multiple hypotheses (FDR < 0.01, s0 = 1, permutations = 250), followed by post-hoc pairwise comparison unpaired t-tests using the same parameters and permutation-based FDR correction (https://cran.r-project.org/web/packages/samr/samr.pdf). Significantly regulated proteins were colored in red and blue in the volcano plots for up and downregulated hits, respectively.

## Western blot

For western blot validation, the elution of proteins was carried out by incubating the beads at 80 °C with 2.5X NuPAGE LDS sample buffer (NP0007, Thermo Fisher Scientific). Samples were loaded on 4–12% NuPAGE Bis–Tris polyacrylamide gel (NP0322, Thermo Fischer Scientific), running with NuPAGE MOPS SDS running buffer (NP0001, Thermo Fischer Scientific) at 110 V for 120 min. Transfer to Immobilon-P 0.45 μm PVDF membrane (Merck Millipore) was carried out in NuPAGE Transfer buffer (NP00061, Thermo Fischer Scientific), supplemented with 10% methanol. Transfer conditions were 120 min with constant voltage at 100 V. Blots were blocked with 5% skim milk solution in PBS-T [1X PBS, 0.05% Tween-20]. Incubations with primary antibodies were carried out overnight as a 1:1,000 dilution. Antibody incubations were followed by washing with PBS-T 3 times for 5–10 min. Secondary antibody against rabbit/mouse IgG was diluted 1:10,000 in PBS-T. Signal was detected with the SuperSignal ECL reagent (34579, Thermo Fischer Scientific) and visualized with GE Amersham Imager 600. Western blot bands were quantified with ImageJ software[71]. Quantified band intensities from pulldowns were normalized to protein levels in input lysates and expressed as pulldown/nuclear extract ratio. Three or four replicate experiments were used for quantification, statistical significance was estimated with paired Student's t-test (*p < 0,05).

## Native RNA-immunoprecipitation

Native RIP for endogenous mRNAs (*VEGFA, MYC, BCL-2, NRAS*) and non-coding RNAs (*RN7SK, GAS5*) was carried out using RUVBL2 and Rabbit IgG1 antibodies. Protein A and G coated Dynabeads (Thermo Fisher) were first washed in RIP lysis buffer containing 25 mM Tris, 150 mM KCl, 0.5% Igepal CA-630, 5 mM DTT, 20 U/ml Rnase inhibitor (Rnasin, Promega), 1X Protease inhibitor cocktail (cOmplete, Roche). Prior to immunoprecipitation, antibodies were incubated with Protein A/G beads, using 4.8 μg of antibodies with 20 μl A+G beads. 20–30 million K562 were harvested for each RNA immunoprecipitation and lysed with 700 μl RIP lysis buffer for 25 min on ice, after which lysates were centrifuged 25 min at 4 degrees centigrade × 17,000 × g. 1% of lysate was removed for input analysis. RNA immunoprecipitation was carried out over 16 h at 4 degrees centigrade with rotation. Beads were washed 5 times 10 min. Protein-bound RNA was eluted by incubating the beads with TRIzol, following RNA extraction. cDNA was prepared from equal volumes of immunoprecipitated and input RNA using Maxima H Minus reverse transcriptase (Thermo Fisher) and random hexamer primers. cDNA was diluted for the detection of RN7SK transcript 1:130. qPCR analysis was carried out using Platinum SYBR Green (Thermo Fisher). CT values were converted by $2^{-CT}$ method and normalized to input levels. Statistical significance was estimated using two-way ANOVA, with Šídák's multiple comparisons test (GraphPad Prism). Statistical significance levels based on p-value: ns = not significant; *p < 0.05; **p < 0.01; ***p < 0.001.

## Primers

VEGFA_fwCTTGCCTTGCTGCTCTACCT
   VEGFA_rvGGTCTCGATTGGATGGCAGT
   MYC_fwCAGGACCCGCTTCTCTGAAA
   MYC_rvTAACGTTGAGGGGCATCGTC

BCL-2_fwGAGAGTGCTGAAGATTGATGGGA
BCL-2_rvTCACGCGGAACACTTGATTCT
NRAS_fwGGGCTGTTCATGGCGGTTCC
NRAS_rvACCACCTGCTCCAACCACCAC
7SK_fw CATCCCCGATAGAGGAGGAC
7SK_rv GCGCAGCTACTCGTATACCC
GAS5_fw CTGTGAGGTATGGTGCTGGG
GAS5_rv AGCTATTCTCATCCTTCCTTGGG

## Antibodies

RUVBL2 (Abcam ab36569)
   NCL (Abcam, ab136649)
   Goat anti-mouse (Thermo Fischer Scientific, G-21040)
   Goat anti-rabbit (Thermo Fischer Scientific, 31460)

***cat*RAPID predictions of protein-RNA interactions.** We employed the original version of *cat*RAPID[30] to predict the G4 interaction propensity of chromatin, K+ and Li+-related proteins. The *cat*RAPID algorithm estimates the interaction through van der Waals, hydrogen bonding, and secondary structure propensities of both protein and RNA sequences[72]. As reported in an analysis of about half a million experimentally validated interactions[73], *cat*RAPID can separate interacting vs. non-interacting pairs with an area under the curve (AUC) receiver operating characteristic (ROC) curve of 0.78 (with false discovery rate (FDR) significantly below 0.25 when the Z-score values are >2). Further information about the method can be found at http://s.tartaglialab.com/page/catrapid_group. In Fig. 1, each protein is segmented into fragments consisting of 50 amino acids, a methodology adapted from previously established protocols[74,75]. The rationale behind dividing proteins into fragments was to account for the bias in *cat*RAPID signal caused by varying sequence lengths. To determine a protein's preference for binding to folded G4A4, we considered two types of "secondary structure occupancy" of RNA: folded (structured) and unfolded (linear) G4A4 (Supplementary Data 1). We classified a protein as a preferential binder of folded G4A4 if more than 75% of the contacts within each 50 amino acid fragment exhibited interaction propensities for folded G4A4 that were higher than those for linear G4A4. In Fig. 4a, we reported the cumulative enrichment of proteins from the K+ group over those from the Li+ group at a specific interaction propensity score. Specifically, the enriched is calculated considering the difference between the catRAPID scores for the folded and unfolded states of G4A4 for each protein (Supplementary Data 2).

***clever*MACHINE classification of protein sets and G4-FUNNIES.** The *clever*MACHINE algorithm contrasts two protein datasets using a combination of distinct physico-chemical properties, including hydrophobicity, structural properties (alpha-helix and beta-sheet, turn), disorder, burial, aggregation, and nucleic acid-binding propensities[33]. This analysis aids in building a model for protein characterization. More insights into the algorithm are available at http://s.tartaglialab.com/page/clever_suite.

In Fig. 4b, c further elaborated in Supplementary Fig. 2, we contrasted the two nuclear protein sets binding to G4 (G4A4). The distinction is based on G4 fold variations when exposed to K+ and Li+, forming the foundation for our method. This G4A4 *clever*Machine model, named 'G4-Folded/UNfolded Nuclear Interaction Explorer System' (G4-FUNNIES), is available at http://service.tartaglialab.com/new_submission/G4FUNNIES to estimate the RNA G4-binding propensities of proteins. Figure 6a illustrates G4-FUNNIES application in differentiating chromatin proteins based on their propensity for G4A4.

In the *clever*MACHINE classification, the three-scale combination (classical[76] and nonclassical[5] RNA-binding ability as well as burial energy[77]) achieved a True Positive Rate (TPR) of 0.99, false positive rate (FPR) of 0.06, and an MCC of 0.907, with the highest cross-validation accuracy of 0.96. The five-scale combination (including

hydrophobicity[78] and aggregation[79]) showed a TPR of 1.00, an FPR of 0.05, and an MCC of 0.928, but a slightly lower cross-validation accuracy of 0.91. Further details on the statistics related to the *clever*MACHINE approach are available at http://service.tartaglialab.com/static_files/algorithms/clever_machine/documentation.html and http://service.tartaglialab.com/static_files/algorithms/clever_machine/tutorial.html.

Before executing G4-FUNNIES on submitted protein sequences, we incorporated two filters: 1) one leveraging the catRAPID signature approach[80] to omit proteins lacking RNA binding capabilities. Further details on this filter can be found in a previously published work[75]; 2) to differentiate between G4-binding and non-G4-binding proteins. In Fig. 5a, b, we introduce the approach to identify G4 vs non-G4-binding proteins based on the LFQ scores from mass spectrometry data: proteins with scores below -15 for K+ and Li+ are deemed as non-G4 binders, while those with scores >= -15 are considered G4 binders (Supplementary Data 2). The two classes have been used to create a *clever*MACHINE model. To validate this classification, we used *pqsfinder*[48] for assessing the G4 RNA affinity of eCLIP proteins[81]. More in detail, we calculated the ratio (G4-noG4)/(G4+noG4), where G4 and noG4 represent the count of G4 RNA within respective groups identified by *clever*MACHINE. This ratio showed a positive correlation with both the pqsfinder score (Fig. 5a) and *clever*MACHINE confidence level (Fig. 5b), validating our initial protein categorization from the mass spectrometry data.

Using the *clever*MACHINE we also analyzed the cytoplasmic interactome of another G4 RNA (G3A2). We built a predictor on the G3A2 dataset distinguishing G4 binding and G binding proteins. We found a convergence in the physicochemical property patterns between the G3A2 and our G4-FUNNIES predictors (see Supplementary Fig. 2). The G3A2 predictor identifies 90% of folded G4 binding proteins and 10% of unfolded ones within the G4A4 dataset. G4-FUNNIES detects 65% of non-G4-binding proteins and 45% of G4-binding proteins within the G3A2 dataset.

All the models generated in these analyses are accessible at http://service.tartaglialab.com/static_files/algorithms/clever_G4_classifier/G4_featured_submissions.html.

**catGRANULE predictions of phase separation.** The tendency of proteins to phase separate (Supplementary Data 6 and 9) is predicted through *catGRANULE*[35]. The algorithm exploits predictions of RNA binding ability and structurally disordered propensities and was employed in our analysis to discriminate protein binding to G4 in the presence of K$^+$ or Li$^+$. Further information can be found at http://s.tartaglialab.com/new_submission/catGRANULE.

**G4 occurrence predictions.** G4 motifs predictions were carried out using *pqsfinder* package (version 2.8.0) in an R (4.1.0) environment[48]. As input to the pipeline we used K$^+$ and Li$^+$ binding sites for Human protein–RNA interactions that were collected from eCLIP experiments[81] with stringent cut-offs [−log10($p$-value) >3 and −log2(fold_enrichment) >3]. *pqsfinder* was used with default parameters and score = 52 (default) was used as the threshold score for accepting the occurrence of a G4 motif.

**AlphaFold predictions of structural disorder.** We used AlphaFold for calculations of protein structures[82]. The PDBs, available from https://alphafold.ebi.ac.uk/, have been analyzed using STRIDE[83]. We counted the amino acids that fall into the categories of Coil and Turn (unstructured elements) as well as Helix and Strand (structured elements). We then determined the fraction of structured elements (Supplementary Data 5).

**Reporting summary**
Further information on research design is available in the Nature Portfolio Reporting Summary linked to this article.

## Data availability
The sequences utilized in the manuscript were obtained from Uniprot, accessible at https://www.uniprot.org/, and their respective structures were sourced from AlphaFold, available at https://alphafold.ebi.ac.uk/. The catalog of phase-separating proteins was taken from https://llps.biocuckoo.cn/index.php. The web service in the manuscript http://service.tartaglialab.com/new_submission/G4FUNNIES is an application of a general algorithm that we previously published[33] and available at http://s.tartaglialab.com/page/clever_suite/. The mass spectrometry proteomics data generated in this study have been deposited to the ProteomeXchange Consortium via the PRIDE repository with dataset identifier PXD041154. Source data are provided in this paper.

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

## Acknowledgements
The mass spectrometry analysis was made by Proteomics Research Infrastructure at the University of Copenhagen. Work in the author's laboratories is supported by the ERC ASTRA_855923 (G.G.T.) and EIC Pathfinder IVBM4PAP_101098989 (G.G.T.), The Novo Nordisk Foundation, Independent Research Fund Denmark, The Lundbeck Foundation, Danish Cancer Society and Carlsberg Foundation (U.A.V.Ø.). The authors would like to thank the RNA initiative and IIT and a grant from the National Center for Gene Therapy and Drugs based on RNA Technology (CN00000041, EPNRRCN3) supported by the European Union.

## Author contributions
J.L. and U.A.V.Ø. conceived the experimental strategy, and J.L. performed all experiments. A.A. and G.G.T. conceived computational analysis strategies and performed computational analysis. All authors contributed to the analysis and interpretation of the data. G.G.T. and U.A.V.Ø. drafted the manuscript, and all authors commented on and accepted the final manuscript. G.G.T. and U.A.V.Ø. secured funding and supervised computational and experimental work.

## Competing interests
The authors declare no competing interests.
