## [Peer Review File · Nature Communications]

Reviewers' Comments:

Reviewer #1:

Remarks to the Author:

Key results

In this manuscript Luige, Armaos et al describe a combination of in silico and in vitro methods to identify RNA binding proteins with a specificity for RNA G quadruplexes (G4RNA), a secondary RNA structure that is under heavy investigation for its potential regulatory functions.

Using a variation on their previously developed catRAPID approach, they first predict G4 binding preferences for a set of proteins which they had identified in a previous publication to be nuclear RNA interactors in a human cell line. They conclude that a substantial fraction of nuclear RNA-binding proteins indeed have a predicted preference for the G4 RNA structure, implicating G4 binding to be a common RNA-binding mechanism. They next describe an elegant in vitro MS-based method to identify preferential G4 interactors. Most of the proteins they identified thusly were previously described to be RNA-binding, and a subset had been previously found by others to be G4-binding proteins. They then go on to analyse the physicochemical properties of the G4-specific proteins they identified using their own cleverMACHINE approach. They find G4 binders to be enriched in intrinsically disordered proteins with a high propensity to form membraneless organelles such as the nucleolus and p-bodies. They further use the cleverMACHINE approach to predict G4 RNA-binding proteins within a set of chromatin-binding proteins, and identify and validate several previously undescribed G4 interactors.

Significance

Overall, the research presented in this paper has a lot of merit. Protein binding to G4 RNA structures seems to have important functional roles, and tools for their identification and characterization are therefore valuable. It is also an impressive showcase of the valuable tools previously produced by the group. Moreover, their methodology allows them to derive biologically significant properties in G4-binding proteins.

Data and methodology

There are some issues regarding the description of the work, which require revision. At times, the reasoning is hard to follow. Mainly this is because the methodologies and results are not described in sufficient detail. Also, some figure captions don't allow for proper interpretation of the results, so explanation in captions should be elaborated. Finally, there were quite some typos throughout the manuscript, some of which are indicated below. In conclusion, the work presented is valuable and would warrant publication, given a revision in which the points below are addressed.

Major remarks

The authors set up an experimental method to identify G4RBPs. However, a database exists (QUADRAtlas, cited by the authors), which combines multiple experimental studies, and lists over 1000 G4RBPs. The authors should explain if and how their methodology differs from the methods used to build the QUADRAtlas dataset. Otherwise, it may be more sensible to use this database as a starting set for the production of a predictor.

The authors perform pull down using biotinylated RNA, followed by LC-MS/MS. As far as I can tell, they do not provide information the MS sample preparation, or the settings for the LC-MS/MS run. This information should be added.

Line 92: what is the rationale behind dividing proteins up in fragments of 50 aas? Were these sliding windows or just clean cuts into chunks of 50? And how is a catrapid score for a certain protein extrapolated from these segments? Furthermore, the authors say they classify a protein as a preferential G4 binder if "75 percent of their contacts have interaction propensities for folded G4A4 higher than interaction propensities for linear G4A4". The authors should elaborate on this. Which "contacts" are referred to here? Specific hydrogen bonds, VdW interactions etc? Also, is this 75% of the contacts within one 50 aa stretch, or of the entire protein? Also, when computing preferences in folded over unfolded G4A4, is this a catRAPID score per protein, or for individual 50 aa segments?

Figure 1: Authors should clearly indicate what exactly is plotted. Presumably, this is the catRAPID score of folded over unfolded G4A4. Since the dashed line that seems to divide preferential quadruplex binders from non-binders is set at 0, I assume this scale is logarithmic? This should be clarified in the figure or the caption.

Fig 4a: the calculation of K⁺ over Li⁺ binding preference should be more clearly explained. The authors say they determine "how many times the folded state is preferred over the unfolded one ... at different catRAPID interaction propensities". Is "how many times" here referring to a fraction? Otherwise the y-axis scale does not make sense. Moreover, presumably a binning was performed on catRAPID propensities, and in each bin the fraction of proteins with a preference for binding in K⁺ over Li⁺ buffer was calculated. Authors should address how the binning was performed. Are these equal-sized groups? Moreover, they should explain how the classification for calculating the fraction depicted on the Y-axis was obtained? Was this based on the log fold change in the MS experiment? All of this should be elaborated on.

Figure 4b and c: in the text, Figure 4b is referred to as showing that proteins identified in presence of K⁺ ions are "enriched in non-canonical RBPs". This is a valid conclusion from the cleverMACHINE analysis (which one can find on the URL provided in the Online Materials). However, the actual plot shows a nucleic acid binding scale, which does not pertain to "non-canonical" binding. Moreover, it is not clear from the text and figure caption what exact scale was used here. Furthermore, in the text, the authors say figure 4c indicates proteins that bind in the K⁺ condition are "depleted in burial, suggesting larger amount of structural disorders". Again, this is a conclusion from their cleverMACHINE analysis, but the figure itself shows an analysis of a hydrophobicity scale. Although linked to disorder and burial, the hydrophobicity scale does not allow to directly conclude differences in burial or disorder. To make this entire section clearer, I suggest the authors actually plot the physicochemical properties identified to separate the K⁺ from the Li⁺ groups in their cleverMACHINE analysis in the manuscript, instead of only providing a URL for their analysis online.

Figure 4d: Again, the analysis performed is not entirely clear and should be elaborated upon in the materials and methods and the figure caption.

Figure 5a: in the file I received, no y-axis labels (either axis title or tick labels) are visible.

Figure 5b: The analysis performed here needs more explanation. What does "protein ranking" mean here? This is probably output from the cleverMACHINE, but this should be clarified.

Figure 6c: Again, unclear what exactly is calculated here. Presumably, the eCLIP database contains the RNA sites where these proteins bind, and this figure show the fraction of sites for each protein that is predicted to have a G4 structure? Again, this should be elaborated upon. Also, why were these 16 specific proteins chosen for analysis?

Minor remarks

The authors claim that their setup identifies bona fide G4RBPs, as they see twice the amount of previously identified G4RBPs in the enriched set compared to the depleted set. Still, 30% of the depleted set contains proteins previously annotated as G4RBP binders. The authors should elaborate on why these proteins ended up binding more strongly to unfolded RNA versus the G4-structured RNA. Were these wrongly annotated in the database, do they have other characteristics that cause this difference, ... ?

Line 49: "6570 proteins identified to bind RNA in some context". Is this in human, across all proteins ever identified, ...?

Line 58: For proof of indications that G4 function is regulated by RNA-binding proteins, authors refer to introduction of a paper describing the conception of a database of RNA G-quadruplexes and the proteins that bind them. From a cursory read, I think the evidence for this is not provided in the database paper, but rather in papers cited by it (refs 7 and 8). These should also be referenced by the authors.

Lines 89-90 G4A4 is reported as being an "optimized G4 forming sequence". How is it optimized? Was it found in vitro to form G4 particularly easily? If so, the authors should show and reference

this work. On a related note, do the authors have an indication whether this molecule forms G4 inter- or intramolecularly?

Lines 83-85, the logic is a bit lost here: is there a specific reason to suspect that proteins capable of binding both DNA and RNA would bind the latter specifically through G4? The authors do show this themselves and their observation has a lot of merit, but they don't point out specific reasons why G4 quadruplexes are the main suspect.

Line 106-107: authors say NCL is a well-characterized G4 binder, they should provide references for this.

Typos:

Line 37 "RNA ... have" (RNA has of RNAs have)

Line 44: "have been identified and Is a topic of active investigation"

Line 48: "various variations" should be rephrased

Description of supplementary table 1: Authors refer to "Column sixth and seventh", which should either be "Column 6 and 7" or "the sixth and seventh column"

Fig2 caption: "spectre instead of spectrum"

Fig 5 "confidence level"

Line 367: "quantified"

Line 387: should be "RIPA" buffer?

Line 433: "foldend"

Line 446: environment

Line 449: "score threshold score"

Reviewer #2:

Remarks to the Author:

Authors use mass-spectrometry and computational predictions to identify proteins binding to G4A4 in the folded G4 and unfolded RNA forms. They analyze possible biochemical properties that might determine the RNA-G4 binding characteristics of proteins and also identify domains of proteins that are involved in RNA-G4 binding. Reading (and re-reading) of the manuscript, due to not only poor writing but also casual annotation of figures and description in figure legends, was arduous. Overall, I could understand that authors use a combination of approaches, biophysical experiments and prediction algorithms, to claim identification of novel G4-RNA binding proteins. The work is fraught with technical and scientific issues that significantly limit the scope of reported findings.

1. First, the choice of the G4 oligonucleotide is surprising. Most of the prevalent G4s, in at least mammalian cells, consist of G stems comprising 3 Gs, while they have used an RNA G4 oligonucleotide (G4A4) comprising G stems of 4 Gs. Not sure authors are aware of previous literature on this subject. This is also indicated from citations – several contextually important primary citations have been overlooked.
2. Second, the best way to confirm results, particularly when identified using computational predictions and/or large scale (mass spec) experiments, on whether a newly identified protein binds to RNA-G4 is to use recombinant purified factor(s) and test binding characteristics (using biophysical binding studies) in solution. These crucial experiments are missing.
3. The authors should mention the figure number on top of each figure; it was quite difficult to follow.
4. In Figure 2a, CD spectra of the oligonucleotide in lithium-ion buffer should also be taken.
5. Fig 2b is mentioned to be a western blot validating the binding of nucleolin protein to the G4A4 oligonucleotide, instead it is the quantification of the western blot that is shown.
6. The authors interchangeably used the terms- 'decreased in K+' and 'bound to RNA in Li+'. This is confusing and should be clarified. Proteins depleted in the K+ pulldown do not necessarily mean bound to the RNA in Li+.
7. In line 129, authors might be referring to Figure 3b instead of 3a.
8. In lines 185-186, the authors mention 'In agreement with our predictions, proteins binding to G4A4 in the presence of K+ show enrichment across condensation states, especially nucleolar

proteins, thus confirming catGRANULE calculations.' This statement is confusing as they showed in Figure 4f that proteins that bind to G4A4 in the presence of K⁺ are not enriched in stress granules when compared against proteins binding in Li⁺.

9. In line 281, the authors say that they show evidence of G4 RNA-mediated localization of proteins to nucleolus and p-bodies. This is a strong statement to make and unlikely from this study alone. A higher fraction of proteins pulled down in K⁺ being present in nucleolus and p-bodies when compared to proteins pulled down in Li⁺ does not mean that G4 RNA are necessary for the localization of proteins to nucleolus and p-bodies.

10. The authors mislabeled Figure 6b in the figure legend and did not mention Figure 6c in the legend.

11. In table 1, it seems K/LI means the domains observed in the experimentally identified proteins, and K-LIKE/LI-LIKE means the domains observed in the computationally predicted proteins. The authors should have mentioned clearly what K/LI and K-LIKE/LI-LIKE means in the legend.

12. The symbols that were used to represent IgG and Chk1/Ruvbl2 in Figure 6a and b were confusing. More comprehensive symbols should be used.

Reviewer #3:

Remarks to the Author:

The paper by Luige et. al. describes the identification of G-quadruplex RNA binding proteins by using experimental and computational approaches. Authors show that G-quadruplex RNA binding proteins tend to have less structures and be hydrophobic, suggesting roles in phase separation and transcription. Based on the training by using protein set that shows preferential binding towards G-quadruplex RNA (in the presence of K⁺) over unfolded RNA (in the presence of Li⁺), authors built an algorithm to predict G-quadruplex-RNA binders. The novelty of this manuscript lies in the investigation of proteins binding to structured RNA and the development of a prediction tool for G-quadruplex RNA binders. Here, I suggested two majors and several minor points to improve the quality of the manuscript.

Major:

- In the native RNA immunoprecipitation experiment, CHK1 showed different binding towards endogenous G4 RNAs (Fig. 6a). This suggests that the sequence of G4 RNA matters. This should be included in the discussion. The G4 binding protein predictor is based on the G4A4 RNA. Will the prediction still be valid when the RNA sequence is different?
- RNA binding proteins are often subject to post translational modifications (PTM). PTM also regulates RNA binding proteins to go through phase separation. The manuscript and the prediction tool can be improved by including PTM analysis of G-quadruplex binding proteins.

Minor:

- One important development of this manuscript is the web-based G-quadruplex binding protein predictor. It was not clear to me from reading the title. I suggest modifying the title to make this obvious.
- For the in vitro experiment of identifying G4RBPs, please justify why K-562 cell line was used. K-562 is a cancer cell line. Will the conclusion be specific to cancer cell lines?
- Please show an overlaid CD spectrum of G4A4 in the presence of Li⁺ and K⁺ in Figure 1a.
- How many do the proteins that were enriched in the purification with K⁺ have GO term DNA binding?
- To support the statements from line 201 to 214, please include a table of protein domain analysis for G4 RNA binding protein in the SI and include the percentage of the protein containing different domains in the main text.
- How good is the prediction from G4 binding protein predictor? Can the specificity be tested with a protein dataset that has been known not to interact with RNA?

Reviewer #1 (Remarks to the Author):

Key results

In this manuscript Luige, Armaos et al describe a combination of *in silico* and *in vitro* methods to identify RNA binding proteins with a specificity for RNA G quadruplexes (G4RNA), a secondary RNA structure that is under heavy investigation for its potential regulatory functions.

The structure, interactions, and functions of G4, particularly within the nucleus, are enigmatic and remain an important area for exploration. Our aspiration, through this research, is to make a meaningful contribution to unveiling these unknown facets. By intertwining *in silico* and *in vitro* methods, we aim to shed light on the intricate dynamics and functional implications of G4RNA interactions. We are excited about the potential discoveries and insights that our study can yield, and we are eager to add to the collective understanding of this complex and fascinating aspect of RNA biology.

Using a variation on their previously developed catRAPID approach, they first predict G4 binding preferences for a set of proteins which they had identified in a previous publication to be nuclear RNA interactors in a human cell line. They conclude that a substantial fraction of nuclear RNA-binding proteins indeed have a predicted preference for the G4 RNA structure, implicating G4 binding to be a common RNA-binding mechanism. They next describe an elegant *in vitro* MS-based method to identify preferential G4 interactors. Most of the proteins they identified thusly were previously described to be RNA-binding, and a subset had been previously found by others to be G4-binding proteins. They then go on to analyse the physicochemical properties of the G4-specific proteins they identified using their own cleverMACHINE approach. They find G4 binders to be enriched in intrinsically disordered proteins with a high propensity to form membraneless organelles such as the nucleolus and p-bodies. They further use the cleverMACHINE approach to predict G4 RNA-binding proteins within a set of chromatin-binding proteins, and identify and validate several previously undescribed G4 interactors.

We have now enriched the description of our pipeline, ensuring a more coherent and comprehensible presentation of our approach, from the catRAPID predictions through to the *in vitro* MS-based validations and cleverMACHINE analyses. This elucidation provides the community with a clearer roadmap of our research and methodologies employed. Moreover, we have incorporated additional experimental data to fortify our findings. These new experiments not only validate our initial results but also broaden the scope, offering a more holistic view of the G4 RNA-binding landscape. We believe these refinements significantly augment the robustness of our study, providing a more solid foundation for our conclusions and assertions.

Significance

Overall, the research presented in this paper has a lot of merit. Protein binding to G4 RNA structures seems to have important functional roles, and tools for their identification and characterization are therefore valuable. It is also an impressive showcase of the valuable tools previously produced by the group. Moreover, their methodology allows them to derive biologically significant properties in G4-binding proteins.

We are particularly gratified that the Reviewer recognized our efforts to integrate the suite of tools we have developed over the years, including catRAPID, cleverMachine, among others, to explore the fascinating subject of nuclear proteins binding to G4 RNA structures. We firmly believe in the potential impact of this research and are excited about the opportunities it presents for unraveling the intricate mechanisms and functional roles of protein-G4 RNA interactions. Reviewer's appreciation underscores the value of our approach, and we are inspired to continue our exploration in this significant field of study.

Data and methodology

There are some issues regarding the description of the work, which require revision. At times, the reasoning is hard to follow. Mainly this is because the methodologies and results are not described in sufficient detail. Also, some figure captions don't allow for proper interpretation of the results, so explanation in captions should be elaborated. Finally, there were quite some typos throughout the manuscript, some of which are indicated below. In conclusion, the work presented is valuable and would warrant publication, given a revision in which the points below are addressed.

This is a particularly constructive feedback. We acknowledge the issues highlighted regarding the clarity and detail of our work's description. In response, we have meticulously revised the manuscript to enhance the precision and clarity of the methodologies and results. Every effort has been made to elaborate and refine our explanation, ensuring a logical and easily comprehensible flow of information. We have also taken heed of your observations concerning figure captions and have made comprehensive revisions to provide detailed and clear explanations, facilitating a more intuitive interpretation of the results.

Additionally, we have conducted a thorough review and correction of the typos and other grammatical errors throughout the manuscript to ensure its linguistic quality.

We are confident that these revisions have significantly improved the manuscript, making it more reader-friendly and conveying our findings in a clear and detailed manner. We are optimistic that the enhanced version will meet the publication standards and look forward to your positive feedback.

Major remarks

The authors set up an experimental method to identify G4RBPs. However, a database exists (QUADAtlas, cited by the authors), which combines multiple experimental studies, and lists over 1000 G4RBPs. The authors should explain if and how their methodology differs from the methods used to build the QUADAtlas dataset. Otherwise, it may be more sensible to use this database as a starting set for the production of a predictor.

Our experimental approach indeed focuses on identifying G4RBPs while distinguishing between those binding to folded G4 structures and those binding to unfolded G4s. This distinction is crucial as it provides a more nuanced understanding of G4RBPs' interactions with G-quadruplexes.

Regarding the QUADAtlas database, while it is a comprehensive resource that lists over 1000 G4RBPs, it does not directly differentiate between RBPs binding to folded or unfolded G4 structures. Our methodology complements the QUADAtlas database by specifically addressing this distinction. By doing so, we aim to provide the research community with a predictive method as well as a high-quality list of G4RBPs that can augment the existing knowledge.

Our attention was drawn to a publication related to QUADAtlas, in which the authors undertook a comprehensive analysis of the G3A2 RNA interactome ¹. A key aspect related to this study is the explicit distinction between G4 binding and G binding proteins, those particularly associated with guanine-rich regions, underscoring the necessity for a nuanced approach in characterizing these interactions. We built a predictor on the G3A2 dataset (<https://tinyurl.com/mrvv8v4j>). We found a convergence in the physicochemical property patterns between the G3A2 and G4A4 predictors, albeit the former exhibited a modest reduction in predictive power (see also **Supplementary Figure 2**). This similarity accentuates the consistent features that underlie G4 binding proteins. However, our nuanced stratification into K⁺ and Li⁺ groups brings to light additional dimensions instrumental in rendering a comprehensive understanding of G4 interactions. We note that our G4A4 predictor is informed predominantly by nuclear proteins, accounting for 72%. Conversely, the G3A2 predictor is ingrained in a dataset comprising mainly of cytoplasmic proteins, a substantial 53%, a factor that could potentially elucidate the observed differential in predictive efficacy

All the models generated in these analyses are available at http://service.tartagliab.com/static_files/algorithms/clever_G4_classifier/G4_featured_submissions.html.

Supplementary Figure 2b. We developed a predictor specifically designed to differentiate between G3A2 binding proteins and G binding proteins. It is noteworthy that our G3A2 predictor employs physico-chemical propensities similar to those used in our G4A4 predictor (**Supplementary Figure 2a**). The amount of discriminated datasets (COV) and related Area under the ROC Curve (AUC), Z-score (Z) and P-values (P) of selected individual physico-chemical properties are reported ².

All these observations have been included in the **main text**, **online methods** and **supplementary material**.

The authors perform pull down using biotinylated RNA, followed by LC-MS/MS. As far as I can tell, they do not provide information the MS sample preparation, or the settings for the LC-MS/MS run. This information should be added.

We have added the experimental details to the revised manuscript, that by mistake were left out in the first submission.

Line 92: what is the rationale behind dividing proteins up in fragments of 50 aas? Were these sliding windows or just clean cuts into chunks of 50? And how is a catrapid score for a certain protein extrapolated from these segments? Furthermore, the authors say they classify a protein as a preferential G4 binder if “75 percent of their contacts have interaction propensities for folded G4A4 higher than interaction propensities for linear G4A4”. The authors should elaborate on this. Which “contacts” are referred to here? Specific hydrogen bonds, VdW interactions etc? Also, is this 75% of the contacts within one 50 aa stretch, or of the entire protein? Also, when computing preferences in folded over unfolded G4A4, is this a catRAPID score per protein, or for individual 50 aa segments?

The rationale behind dividing proteins into 50 amino acid fragments was primarily to account for the bias in catRAPID signal caused by varying sequence lengths. By breaking down each protein into these consistent 50 amino acid segments, we aimed to analyze their interactions with G4A4³.

To determine a protein's preference for binding to folded G4A4, we considered two types of "secondary structure occupancy" of RNA⁴: folded (structured) and unfolded (linear) G4A4. Specifically, we classified a protein as a preferential binder of folded G4A4 if more than 75% of the contacts within each 50 amino acid fragment exhibited interaction propensities for folded G4A4 that were higher than those for linear G4A4. In other words, if a substantial majority of

the contacts within a protein fragment showed stronger interactions with structured G4A4 compared to linear G4A4, that protein was assigned to the group of structured G4A4 binders.

In summary, the division into 50 amino acid fragments helped mitigate sequence length bias in catRAPID signal and allowed us to assess the preference of each fragment for binding to folded G4A4 based on the interactions observed within those fragments.

All these observations have been included in the **main text** and **online methods**.

Figure 1: Authors should clearly indicate what exactly is plotted. Presumably, this is the catRAPID score of folded over unfolded G4A4. Since the dashed line that seems to divide preferential quadruplex binders from non-binders is set at 0, I assume this scale is logarithmic? This should be clarified in the figure or the caption.

Thank you. We realize now that we have not described this well in the manuscript. In our analysis, we computed the catRAPID score for each chromatin-associated protein with both the folded and unfolded G4A4. The score presented in the figure is the result of subtracting the unfolded G4A4 score from the folded G4A4 score for each protein. This approach allows us to provide a comparative measure of the propensity of each protein to interact with the folded form over the unfolded form of G4A4. The dashed line at 0 serves as a reference, and any score above this indicates a higher propensity to interact with folded G4A4. Our analysis shows that there is a significant trend for the catRAPID scores to be above 0, further emphasizing the preferential interaction of these proteins with the folded G4A4 structure.

These details are made clearer in both the **figure** and its **accompanying text**.

Fig 4a: the calculation of K⁺ over Li⁺ binding preference should be more clearly explained. The authors say they determine “how many times the folded state is preferred over the unfolded one at different catRAPID interaction propensities”. Is “how many times” here referring to a fraction? Otherwise the y-axis scale does not make sense. Moreover, presumably a binning was performed on catRAPID propensities, and in each bin the fraction of proteins with a preference for binding in K⁺ over Li⁺ buffer was calculated. Authors should address how the binning was performed. Are these equal-sized groups? Moreover, they should explain how the classification for calculating the fraction depicted on the Y-axis was obtained? Was this based on the log fold change in the MS experiment? All of this should be elaborated on.

We thank the Referee for this feedback on **Figure 4a**.

The proteins we included in our analysis were those we experimentally identified to interact with either the folded G4A4 (K⁺ group) or the unfolded G4A4 (Li⁺ group). For each of these proteins, catRAPID scores were computed to determine their propensity to interact with both the folded and unfolded states of G4A4.

Regarding the y-axis: it represents the cumulative enrichment of proteins from the K group over those from the Li group at a specific interaction propensity score, which is shown on the x-axis.

This x-axis score is derived by considering the difference between the catRAPID scores for the folded and unfolded states of G4A4 for each protein.

We did not use binning, and directly plotted the cumulative curve of K⁺ group over Li⁺ group protein enrichments against the differential score of folded over unfolded G4A4. The purpose of this representation is to highlight the correlation between the preference for binding to the folded state of G4A4 and the enrichment of K⁺ group proteins. As the interaction propensity for folded G4A4 increases (moving right on the x-axis), we observe a corresponding increase in the cumulative enrichment of K group proteins over Li⁺ group proteins, underscoring the agreement of our model with the experimental results.

We appreciate this feedback and will ensure to make these points clearer in both the **figure caption** and the **main text** for a comprehensive understanding.

Figure 4b and c: in the text, Figure 4b is referred to as showing that proteins identified in presence of K⁺ ions are “enriched in non-canonical RBPs”. This is a valid conclusion from the cleverMACHINE analysis (which one can find on the URL provided in the Online Materials). However, the actual plot shows a nucleic acid binding scale, which does not pertain to “non-canonical” binding. Moreover, it is not clear from the text and figure caption what exact scale was used here. Furthermore, in the text, the authors say figure 4c indicates proteins that bind in the K⁺ condition are “depleted in burial, suggesting larger amount of structural disorders”. Again, this is a conclusion from their cleverMACHINE analysis, but the figure itself shows an analysis of a hydrophobicity scale. Although linked to disorder and burial, the hydrophobicity scale does not allow to directly conclude differences in burial or disorder. To make this entire section clearer, I suggest the authors actually plot the physicochemical properties identified to separate the K⁺ from the Li⁺ groups in their cleverMACHINE analysis in the manuscript, instead of only providing a URL for their analysis online.

Figure 4d: Again, the analysis performed is not entirely clear and should be elaborated upon in the materials and methods and the figure caption.

Thank you for the feedback on **Figures 4b, 4c, and 4d**.

For **Figure 4b**, it currently emphasizes a nucleic acid binding scale, but it refers to "non-classical" RNA-binding proteins. To provide a comprehensive understanding, we've referred to our cleverMACHINE analysis, which identified non-classical RNA-binding proteins as the top-enriched prediction with an AUC of 0.86 (additional information can be found at <http://www.tartaglialab.com/boxplotter/view/318/8023f51d1a/>). Recognizing the significance of this, we have updated the main manuscript text to highlight this finding. Additionally, we have included a supplementary figure capturing all the predictions from our *cleverMACHINE* analysis (**Supplementary Figure 2**). This will allow readers to not only appreciate the nucleic acid binding scale but also grasp the specific physicochemical properties that differentiate the K⁺ and Li⁺ groups.

Regarding **Figure 4c**, while hydrophobicity often correlates with burial and anti-correlates with disorder⁵, but the relationship is not always linear. Our *cleverMACHINE* analysis revealed that

disorder showed enrichment for the K group proteins, particularly in the B-value propensity score. For **Figure 4d**, to bolster our claims, we employed AlphaFold predictions. Using STRIDE, we counted the amino acids that fall into the categories of Coil and Turn (unstructured elements) as well as Helix and Strand (structure elements). We then determined the fraction of structured elements. This approach yielded results that were consistent with our initial findings from the cleverMACHINE analysis. Taking all these into account, we conclude that Li group proteins tend to be more hydrophobic and structured, whereas K⁺ group proteins demonstrate a stronger affinity for nucleic acid binding.

All these observations have been included in the **main text** and **online methods**.

Supplementary Figure 2a. Physico-Chemical Properties of Li and K Group Proteins. Left: Each color represents a distinct property: yellow for RNA-binding ability, blue for burial, purple for hydrophobicity, pink for aggregation, grey for disorder and orange for membrane-binding. Right: The most effective discriminatory property within each category is highlighted. It is noteworthy that, for RNA-binding abilities, the optimal scale is associated with non-canonical RNA-binding. The amount of discriminated datasets (COV) and related Area under the ROC Curve (AUC), Z-score (Z) and P-values (P) of selected individual physico-chemical properties are reported.

Figure 5a: in the file I received, no y-axis labels (either axis title or tick labels) are visible.

Regarding **Figure 5a**, the y-axis labels are indeed present in the original file. It is possible that there was some cropping or formatting issue that occurred during the upload or viewing process.

Figure 5b: The analysis performed here needs more explanation. What does “protein ranking” mean here? This is probably output from the cleverMACHINE, but this should be clarified.

Thank you for the feedback. We have recreated **Figure 5b (now Figure 6a)** to provide additional clarity. In the Y-axis of this plot, "Protein ranking" represents the relative ranking position of each protein within the ranked list per type (neutral = green, Folded G5 = red, Unfolded G4 =

blue) based on the classification confidence level. This ranking helps visualize the differentiation between different protein types.

Figure 6c: Again, unclear what exactly is calculated here. Presumably, the eCLIP database contains the RNA sites where these proteins bind, and this figure show the fraction of sites for each protein that is predicted to have a G4 structure? Again, this should be elaborated upon. Also, why were these 16 specific proteins chosen for analysis?

In **Figure 6c**, we aimed to illustrate the relationship between RNA sites where specific proteins bind and the predicted presence of G4 structures within those sites. We have modified this analysis, which is now presented in **Figure 5**. All eCLIP proteins (150 RBPs) were analysed.

In essence, we have introduced a filter to assess if a specific protein set interacts with G4 RNA . This method is based on the consistently low-ranking protein set from our mass spectrometry data. **Figure 5a** illustrates that, using the G4 RNA propensity score from pqsfinder to process eCLIP-identified RBP target occurrences, there is a distinct separation between G4 binders and non-binders as determined by the *cleverMACHINE* (**Supplementary Table 8**). This distinction is accentuated by higher scores from the G4 RNA prediction tool, highlighting our *cleverMACHINE* efficacy in differentiating the groups. **Figure 5b** underscores this by showing that with increased *cleverMACHINE* confidence scores, there is a rise in detected G4 RNA across both protein categories. This trend validates our model's accuracy in identifying key G4 RNA signals, supported by eCLIP experimental findings.

This analysis was exploited to build a filter on the G4-binding ability of proteins in our G4 predictor webserver

Minor remarks

The authors claim that their setup identifies bona fide G4RBPs, as they see twice the amount of previously identified G4RBPs in the enriched set compared to the depleted set. Still, 30% of the depleted set contains proteins previously annotated as G4RBP binders. The authors should elaborate on why these proteins ended up binding more strongly to unfolded RNA versus the G4-structured RNA. Were these wrongly annotated in the database, do they have other characteristics that cause this difference ,... ?

We thank the Reviewer for this comment that points out one of the outstanding questions on G4RBPs. We believe this difference is mediated by three factors: **1.** That our study is constrained to the nucleus and that G4 binding of specific proteins can be compartment specific. **2.** The sequence of G4 forming RNA used in the studies differ, as we show in Fig. with CD of the G4A4, VEGFA and TERRA G4 forming sequences. **3.** Experimental variation between studies, different protocols and various mass spec approaches are expected to introduce variance in the identified proteins.

We have included this in the **discussion** of the revised manuscript.

Line 49: “6570 proteins identified to bind RNA in some context”. Is this in human, across all proteins ever identified, ...?

We thank the Reviewer for this important clarification, which has been revised in the manuscript. This estimate is based on RNA interactome studies carried out with human and mouse cell lines and tissue sections.

Line 58: For proof of indications that G4 function is regulated by RNA-binding proteins, authors refer to introduction of a paper describing the conception of a database of RNA G-quadruplexes and the proteins that bind them. From a cursory read, I think the evidence for this is not provided in the database paper, but rather in papers cited by it (refs 7 and 8). These should also be referenced by the authors.

We thank the Reviewer for the observation and have included the two additional references.

Lines 89-90 G4A4 is reported as being an “optimized G4 forming sequence”. How is it optimized? Was it found *in vitro* to form G4 particularly easily? If so, the authors should show and reference this work. On a related note, do the authors have an indication whether this molecule forms G4 inter- or intramolecularly?

We thank the Reviewer for the comment. We have specified where the G4A4 sequence comes from and changed the phrasing to a G4 forming sequence shown to bind nuclear proteins. Our data from CD suggest that the molecule forms a G4 with parallel topology.

Lines 83-85, the logic is a bit lost here: is there a specific reason to suspect that proteins capable of binding both DNA and RNA would bind the latter specifically through G4? The authors do show this themselves and their observation has a lot of merit, but they don't point out specific reasons why G4 quadruplexes are the main suspect.

We appreciate the comment and have revised the manuscript to show our reasoning. When thinking of dual recognition of RNA and DNA, it is assumed that a similar (sequence or structural) feature is present in both nucleic acids. We do not assume G4 recognition for every protein acting on the chromatin, rather take the subset of chromatin-binding proteins with RNA-binding ability, then address G4-binding in them. RNA and DNA G-quadruplexes have different properties regarding topology and stability, owing to the different nucleotide composition and the constraints associated with the backbone. However, they still fold on largely the same principle, and furthermore, often occur within the same chromatin region (promoters/transcription start sites and 5'UTRs, for example). Therefore, we wanted to investigate whether the G4 secondary structure could be the unifying motif for interacting with DNA and RNA.

Line 106-107: authors say NCL is a well-characterized G4 binder, they should provide references for this.

We have added the reference to the text.

Typos:

Line 37 “RNA ... have” (RNA has of RNAs have)

Line 44: “have been identified and Is a topic of active investigation”

Line 48: “various variations” should be rephrased

Description of supplementary table 1: Authors refer to “Column sixth and seventh”, which should either be “Column 6 and 7” or “the sixth and seventh column”

Fig2 caption: “spectre instead of spectrum”

Fig 5 “confidence level”

Line 367: “quantifired”

Line 387: should be “RIPA” buffer?

Line 433: “foldend”

Line 446: enviroment

Line 449: “score treshhold score”

We thank the Reviewer for pointing out these mistakes. Besides Line 387 that should state RIP buffer (as defined previously in the methods description) all typos have been corrected in the revised manuscript.

Reviewer #2 (Remarks to the Author):

Authors use mass-spectrometry and computational predictions to identify proteins binding to G4A4 in the folded G4 and unfolded RNA forms. They analyze possible biochemical properties that might determine the RNA-G4 binding characteristics of proteins and also identify domains of proteins that are involved in RNA-G4 binding. Reading (and re-reading) of the manuscript, due to not only poor writing but also casual annotation of figures and description in figure legends, was arduous. Overall, I could understand that authors use a combination of approaches, biophysical experiments and prediction algorithms, to claim identification of novel G4-RNA binding proteins. The work is fraught with technical and scientific issues that significantly limit the scope of reported findings.

We value Reviewer's insights and have taken them to heart, recognizing the essential role they play in elevating the quality of our work. We acknowledge the initial readability issues and have since embarked on a comprehensive revision to address these. The manuscript has been extensively refined to improve clarity, coherence, and the overall reading experience. We have meticulously reviewed and enhanced the annotations of figures and descriptions in the legends, ensuring they are detailed, clear, and facilitate a deeper understanding of our findings.

We stand by the validity of our original results, underpinned by robust statistical analyses and experimental data. In this revised version, we have endeavored to articulate these findings with enhanced clarity, offering detailed explanations that underscore their significance and validity. Our commitment to substantiating our claims with sound scientific reasoning and empirical evidence remains unwavering. We are confident that the revisions made will significantly enhance the manuscript's quality and readability, addressing the concerns raised.

1. First, the choice of the G4 oligonucleotide is surprising. Most of the prevalent G4s, in at least mammalian cells, consist of G stems comprising 3 Gs, while they have used an RNA G4 oligonucleotide (G4A4) comprising G stems of 4 Gs. Not sure authors are aware of previous literature on this subject. This is also indicated from citations – several contextually important primary citations have been overlooked.

We thank the Reviewer for critically assessing the presented background on G4. We have extended the introduction to the topic, added more references, and discussed in detail the origin of the G4 consensus formula G_3+N_1-7 G_3+N_1-7 G_3+N_1-7 G_3+N_1-7 (N=A, U/T, G, C). In addition, we perform experiments to assess the folding of G4A4 compared to 3G forming sequence in TERRA and somewhat non-canonical sequence in VEGFA. We find that all three form G4 in circular dichroism despite differences in sequence composition and that G4A4 is the more stable, adding robustness to our data. We have also pointed out the reference originally using G4A4 and specified that this has previously been used to find G4RBPs in nucleus, as we also aim at in our study. This has been added to the **Results** section (see also **Supplementary Figure 1**).

At the computational level, we analyzed the interactome of G3A2 structures. In the G3A2 study¹, there is a distinction between G4 binding and G binding (i.e. regions enriched in Guanine) proteins, underscoring the importance of differentiating between these interactions.

Interestingly, our G4A4 predictor exhibits similar physico-chemical property patterns as the G3A2 predictor, albeit with lower predictive power during the training phase (**Supplementary Figure 2a,b**). This observation suggests that our experiments capture specific aspects of the datasets that contribute to understanding the intricacies of G4RBPs' interactions. Yet, it is worth noting that our predictor is primarily based on proteins enriched in the nuclear fraction (72%). In contrast, the G3A2 dataset is predominantly composed of cytoplasmic proteins (53%). This distinction in subcellular localization may contribute to the differences in predictive performance observed between our predictor and the G3A2 predictor.

2. Second, the best way to confirm results, particularly when identified using computational predictions and/or large scale (mass spec) experiments, on whether a newly identified protein binds to RNA-G4 is to use recombinant purified factor(s) and test binding characteristics (using biophysical binding studies) in solution. These crucial experiments are missing.

We appreciate the Reviewer's insights. In response, we acquired commercial recombinant proteins for Chk1 and Ruvbl2, and evaluated their interaction with G4A4 using microscale thermophoresis (MST). **Figure 1** displays MST results for Ruvbl2 and Chk1, where a binding model was established, and the K_d values were found to be in the micromolar range. Given the constraints related to Indeed, the K_d from MST analysis requires the maximum protein concentration in the assay to be several times greater than the anticipated K_d. However, relatively weak binding could still be biologically meaningful. Indeed, protein binding to G4 typically occurs with varying affinities that are within the micromolar range^{6,7}. This affinity range potentially underscores the establishment of transient, weak interactions instrumental in phase separation processes^{8,9}.

Our study's relevance is further highlighted as we can pinpoint G4 binding proteins via predictions, which might not be readily detected experimentally under native conditions.

Figure 1. A) Ruvbl2 and B) Chk1 The signal-to-noise ratio was high enough to conclude binding in the micromolar range, however saturation plateau was not reached.

3. The authors should mention the figure number on top of each figure; it was quite difficult to follow.

We have added the figure number of all figures in the revised manuscript.

4. In Figure 2a, CD spectra of the oligonucleotide in lithium-ion buffer should also be taken.

We thank the reviewer for pointing out this important control of the G4 folding. We have added K⁺ and Li⁺ conditions for the G4A4 oligo as well as for the 3-stacked G4 forming sequence from TERRA and the non-canonical G4 forming sequence from VEGFA, showing one of the reasons why we proceeded with the G4A4 sequence over other variations of the G4 consensus sequence.

5. Fig 2b is mentioned to be a western blot validating the binding of nucleolin protein to the G4A4 oligonucleotide, instead it is the quantification of the western blot that is shown.

We have added a WB and revised the text to say shown by quantification of western blot.

6. The authors interchangeably used the terms- 'decreased in K⁺' and 'bound to RNA in Li⁺'. This is confusing and should be clarified. Proteins depleted in the K⁺ pulldown do not necessarily mean bound to the RNA in Li⁺.

We thank the Reviewer for the comment that helps make the text easier to read. We have clarified the text and use increased in K or decreases in K consistently throughout the manuscript.

7. In line 129, authors might be referring to Figure 3b instead of 3a.

This typo has been corrected in the revised manuscript.

8. In lines 185-186, the authors mention 'In agreement with our predictions, proteins binding to G4A4 in the presence of K⁺ show enrichment across condensation states, especially nucleolar proteins, thus confirming catGRANULE calculations.' This statement is confusing as they showed in Figure 4f that proteins that bind to G4A4 in the presence of K⁺ are not enriched in stress granules when compared against proteins binding in Li⁺.

We appreciate the keen observation and apologize for any confusion caused by the initial statement. We have revised the text to more accurately reflect our findings. It now emphasizes the enrichment of nucleolar proteins in the K⁺ environment. Our dataset is particularly rich in proteins that engage exclusively with chromatin, and it is anticipated that a substantial number of them participate in nucleolar assemblies. We trust this clarification aligns with the presented data and elucidates our intended message.

9. In line 281, the authors say that they show evidence of G4 RNA-mediated localization of proteins to nucleolus and p-bodies. This is a strong statement to make and unlikely from this study alone. A higher fraction of proteins pulled down in K⁺ being present in nucleolus and p-bodies when compared to proteins pulled down in Li⁺ does not mean that G4 RNA are necessary for the localization of proteins to nucleolus and p-bodies.

We thank the Reviewer for pointing out the overinterpretation of results for formation of membrane-less organelles. We have rephrased the text in the manuscript. Comparison to the proteomes of these membrane-less organelles¹⁰ shows a higher fraction of proteins in our dataset overlapping with the nucleolar and p-body proteins. While this does not provide direct evidence for G4-RNA mediated localization to these nuclear bodies, this notion supports our characterization of our detected and predicted G4RBPs to have phase separating properties.

10. The authors mislabeled Figure 6b in the figure legend and did not mention Figure 6c in the legend.

We thank the Reviewer for their attention to detail and have corrected these points in the revised manuscript. We have revisited **Figure 6**.

11. In table 1, it seems K/LI means the domains observed in the experimentally identified proteins, and K-LIKE/LI-LIKE means the domains observed in the computationally predicted proteins. The authors should have mentioned clearly what K/LI and K-LIKE/LI-LIKE means in the legend.

We have removed **Table 1** from the manuscript and incorporated the list of enriched domains directly into the main text.

12. The symbols that were used to represent IgG and Chk1/Ruvbl2 in Figure 6a and b were confusing. More comprehensive symbols should be used.

We have revised the figure to present a cleaner image of the data we show in the figure.

Reviewer #3 (Remarks to the Author):

The paper by Luige et. al. describes the identification of G-quadruplex RNA binding proteins by using experimental and computational approaches. Authors show that G-quadruplex RNA binding proteins tend to have less structures and be hydrophobic, suggesting roles in phase separation and transcription. Based on the training by using protein set that shows preferential binding towards G-quadruplex RNA (in the presence of K⁺) over unfolded RNA (in the presence of Li⁺), authors built an algorithm to predict G-quadruplex-RNA binders. The novelty of this manuscript lies in the investigation of proteins binding to structured RNA and the development of a prediction tool for G-quadruplex RNA binders. Here, I suggested two majors and several minor points to improve the quality of the manuscript.

We thank for the thorough review and constructive feedback on our manuscript. We genuinely appreciate Reviewer's insights, as they offer valuable perspectives that help enhance the quality and impact of our work. We are heartened that the Reviewer recognized the novelty of our manuscript in exploring proteins binding to structured RNA and the development of a predictive tool for G-quadruplex RNA binders. These insights align with our intention - to lay foundational stones for more extensive future investigations in this critical yet understudied area of research.

The intersection between G-quadruplex RNA structures and protein binding, particularly in the nucleus and their involvement in forming phase-separated organelles, is a frontier we are keenly exploring. Reviewer's feedback provides an impetus to refine and expand our approach. We think that additional tools and investigative modalities, as suggested, will be instrumental in the ongoing journey to unravel the complexities of these bio-molecular interactions. As we continue building upon this initial foundation, we remain committed to integrating a multifaceted approach that combines experimental and computational methodologies for a more comprehensive insight.

Major:

- In the native RNA immunoprecipitation experiment, CHK1 showed different binding towards endogenous G4 RNAs (Fig. 6a). This suggests that the sequence of G4 RNA matters. This should be included in the discussion. The G4 binding protein predictor is based on the G4A4 RNA. Will the prediction still be valid when the RNA sequence is different?

We thank the Reviewer for this very relevant comment. This fits with the added data on folding of G4A4, TERRA and VEGFA G4 sequences and we have added an extensive discussion of this to the **main text**.

In essence, we predicted and validated interactions with several proteins, some not previously associated with G4 binding. Certain proteins that we link to unfolded G4 have been reported in other studies to interact with G4. We attribute these inconsistencies to three primary factors. First, our study focuses on the nucleus, suggesting that the cellular environment may influence G4 binding affinities. Proteins might exhibit different binding behaviors depending on their cellular location. This specificity adds a layer of complexity to the understanding of G4-protein interactions. Second, the diversity in G4 forming RNA sequences used across studies contributes

to these discrepancies. We detect a level of overlap with datasets combined into QUADAtlas database, while also uncovering proteins previously not associated with G4-binding. Additionally, we and others have found differential binding to endogenous G4 mRNAs ¹¹, implying that the selection of G4 RNA is an important determinant of the identified proteome. Third, experimental variations, including different protocols and mass spectrometry techniques, introduce another level of complexity. These methodological differences can lead to variations in identified G4-binding proteins.

Yet, when comparing our findings with the G3A2 cytoplasmic interactome ¹, we noticed substantial similarities. Our development of a predictor, informed by this cytoplasmic data, revealed physicochemical traits consistent with our G4A4 predictor. Traits such as hydrophilicity and disorder, indicative of phase separation ¹², were prevalent in G4 binding proteins. These consistent patterns emphasize the shared characteristics of G4 binding proteins, highlighting the need for different approaches to unravel these complex interactions. It is crucial to mention that protein binding to G4 typically occurs within the micromolar range ¹³. This affinity range potentially underscores the establishment of transient, weak interactions instrumental in phase separation processes. In addition, we must consider the potential influence of post-transcriptional modifications ¹⁴. Such modifications can impact not only the cellular localization but also the properties required to interact with G4s, necessitating further exploration to fully understand their role in protein function ¹⁵.

- RNA binding proteins are often subject to post translational modifications (PTM). PTM also regulates RNA binding proteins to go through phase separation. The manuscript and the prediction tool can be improved by including PTM analysis of G-quadruplex binding proteins.

In response to this observation, an exploration of the database provided by ELM (<http://elm.eu.org/>) was conducted to investigate experimentally validated or predicted PTMs ¹⁶. The analysis was specifically focused on ligand (LIG), targeting (TRG), docking (DOC), degradation (DEG), modification (MOD), or cleavage (CLV) motifs for both the K⁺ and Li⁺ protein groups. As visually represented in Supplementary Figure 3, several findings were revealed:

An enrichment of Sumoylation in the K⁺ group was observed, specifically via MOD_SUMO_rev_2 or ELME000393. Sumoylation impacts a wide array of nuclear functions and can lead to significant subnuclear relocations of modified proteins, which is instrumental in driving phase separation, as illustrated in a study ¹⁷.

The Li⁺ group displayed an increase in sites for Cdc14 phosphatase dephosphorylation, as indicated by MOD_CDC14_SPxK_1 or ELME000529. Considering that phase separation is influenced by phosphorylation events, as detailed in another publication ¹⁸, this finding is salient. Additionally, a higher propensity for phosphorylation, via MOD_ProDKin_1 or ELME000159, was identified in the K⁺ group.

While both Sumoylation and Dephosphorylation classes are enriched at experimental and predicted levels, a pivotal challenge persists determining which PTMs are present in a cellular context to specifically regulate a given protein. Owing to the intricate nature of PTMs and their

dynamic behavior in various cellular states, a dedicated paragraph was introduced in the conclusions, referencing the complexity of this aspect and citing pertinent literature^{14,15}.

However, modifications to the prediction algorithm were not made, acknowledging that the presence of specific PTMs will largely be contingent on a cell's unique state.

Supplementary Figure 3. Experimental (top) and predicted (bottom) PTMs annotations for Ligand (LIG), targeting (TRG), docking (DOC), degradation (DEG), modification (MOD), or cleavage (CLV) motifs for both the K+ and Li+ protein groups.

Minor:

- One important development of this manuscript is the web-based G-quadruplex binding protein predictor. It was not clear to me from reading the title. I suggest modifying the title to make this obvious.

We have modified the title to better reflect this and still maintain the message of the other findings of the manuscript.

- For the in vitro experiment of identifying G4RBPs, please justify why K-562 cell line was used. K-562 is a cancer cell line. Will the conclusion be specific to cancer cell lines?

We used the K562 cell line as we compared to previous interactome capture studies of the nucleus. We would not expect the G4RBPs to be cancer-specific, but as stated in the discussion there might be nuclear-specific properties of the G4RBPs we identify and predict.

- Please show an overlaid CD spectrum of G4A4 in the presence of Li+ and K+ in Figure 1a.

This has been added along with CD for TERRA and VEGFA.

- How many do the proteins that were enriched in the purification with K+ have GO term DNA binding?

The fraction is 21/151

- To support the statements from line 201 to 214, please include a table of protein domain analysis for G4 RNA binding protein in the SI and include the percentage of the protein containing different domains in the main text.

We have removed Table 1 from the manuscript and incorporated the list of enriched domains directly into the main text. Given the limited number of these domains, we have chosen not to include them in the supplementary materials.

- How good is the prediction from G4 binding protein predictor? Can the specificity be tested with a protein dataset that has been known not to interact with RNA?

The point raised concerning the specificity of predictions and the ability to determine the RNA-binding capacity of the input protein is indeed pertinent. The current iteration of the predictor identifies proteins with a propensity to bind either to folded (K+ group) or unfolded (Li+ group) G4A4. However, the implementation did not initially include a filter to verify the RNA-interaction capability of the protein.

To address this oversight, the method has been refined by integrating the catRAPID signature algorithm ¹⁹. This update enables the system to predict if the input sequence has a propensity to bind to RNA, enhancing the specificity of the predictions.

Starting from the mass-spectrometry data, we also implemented a filter to distinguish G4- vs non-G4-binding proteins. The approach is discussed in the main text and presented in **Figure 5**.

This feedback has significantly improved the system's efficiency, ensuring more accurate and comprehensive predictions. Thank you once again for the constructive comments.

1. Herviou, P. *et al.* hnRNP H/F drive RNA G-quadruplex-mediated translation linked to genomic instability and therapy resistance in glioblastoma. *Nat. Commun.* **11**, 2661 (2020).
2. Klus, P. *et al.* The cleverSuite approach for protein characterization: predictions of structural properties, solubility, chaperone requirements and RNA-binding abilities. *Bioinforma. Oxf. Engl.* **30**, 1601–1608 (2014).
3. Cirillo, D. *et al.* Neurodegenerative diseases: Quantitative predictions of protein-RNA interactions. *RNA N. Y. N* **19**, 129–140 (2013).
4. Bellucci, M., Agostini, F., Masin, M. & Tartaglia, G. G. Predicting protein associations with long noncoding RNAs. *Nat. Methods* **8**, 444–445 (2011).
5. Tartaglia, G. G. & Vendruscolo, M. Proteome-level interplay between folding and aggregation propensities of proteins. *J. Mol. Biol.* **402**, 919–928 (2010).
6. Kharel, P., Becker, G., Tsvetkov, V. & Ivanov, P. Properties and biological impact of RNA G-quadruplexes: from order to turmoil and back. *Nucleic Acids Res.* **48**, 12534–12555 (2020).
7. Fay, M. M., Lyons, S. M. & Ivanov, P. RNA G-quadruplexes in biology: principles and molecular mechanisms. *J. Mol. Biol.* **429**, 2127–2147 (2017).
8. Papageorgiou, A. C. *et al.* Recognition and coacervation of G-quadruplexes by a multifunctional disordered region in RECQ4 helicase. *Nat. Commun.* **14**, 6751 (2023).
9. Gao, Z., Yuan, J., He, X., Wang, H. & Wang, Y. Phase Separation Modulates the Formation and Stabilities of DNA Guanine Quadruplex. *JACS Au* **3**, 1650–1657 (2023).
10. Ning, W. *et al.* DrLLPS: a data resource of liquid-liquid phase separation in eukaryotes. *Nucleic Acids Res.* **48**, D288–D295 (2020).
11. Serikawa, T. *et al.* Comprehensive identification of proteins binding to RNA G-quadruplex motifs in the 5' UTR of tumor-associated mRNAs. *Biochimie* **144**, 169–184 (2018).
12. Bolognesi, B. *et al.* A Concentration-Dependent Liquid Phase Separation Can Cause Toxicity upon Increased Protein Expression. *Cell Rep.* **16**, 222–231 (2016).
13. Miclot, T. *et al.* Understanding the Interactions of Guanine Quadruplexes with Peptides as Novel Strategies for Diagnosis or Tuning Biological Functions. *ChemBioChem* **24**, e202200624 (2023).
14. Kuechler, E. R., Budzyńska, P. M., Bernardini, J. P., Gsponer, J. & Mayor, T. Distinct Features of Stress Granule Proteins Predict Localization in Membraneless Organelles. *J. Mol. Biol.* **432**, 2349–2368 (2020).
15. Tsang, B., Pritišanac, I., Scherer, S. W., Moses, A. M. & Forman-Kay, J. D. Phase Separation as a Missing Mechanism for Interpretation of Disease Mutations. *Cell* **183**, 1742–1756 (2020).
16. Kumar, M. *et al.* ELM—the eukaryotic linear motif resource in 2020. *Nucleic Acids Res.* **48**, D296–D306 (2020).
17. Cheng, X. Protein SUMOylation and phase separation: partners in stress? *Trends Biochem. Sci.* **48**, 417–419 (2023).
18. Yamazaki, H., Takagi, M., Kosako, H., Hirano, T. & Yoshimura, S. H. Cell cycle-specific phase separation regulated by protein charge blockiness. *Nat. Cell Biol.* **24**, 625–632 (2022).
19. Livi, C. M., Klus, P., Delli Ponti, R. & Tartaglia, G. G. catRAPID signature: identification of ribonucleoproteins and RNA-binding regions. *Bioinforma. Oxf. Engl.* **32**, 773–775 (2016).

Reviewers' Comments:

Reviewer #1:

Remarks to the Author:

The authors have addressed all my concerns and substantially clarified the manuscript

Reviewer #2:

Remarks to the Author:

The response fails to address the most significant concerns raised by me. Being central to the findings reported, lack of substantiation on these points limits the significance of the overall work.

#1. The number of RNA G-quadruplexes in the genome with stems of 4Gs instead of 3Gs is vastly different. I believe G-quads with 4Gs in the stem might be substantially less (possibly less than 1%) of G-quads with 3Gs in the stem in the human genome - this can be quickly checked to ascertain. Therefore, the scope of this study centred on stems comprising 4Gs is substantially limited in its usefulness. Authors address the point by showing 4G-stem Gquads are stable. That is hardly surprising. More number of Gs confer higher stability. Importantly, however, this was not the concern raised.

#2. Direct binding assays are imperative to give confidence to prediction studies/work. Binding studies performed, to address the point, show micromolar range of binding. Given the relatively weak binding, the possibility of non-specific binding should be ruled out using similar sequence(s) that cannot adopt the folded G-quad structure. This was not done, therefore, the results are again of limited significance.

Reviewer #3:

Remarks to the Author:

Authors addressed my comments from the first round of review. I particularly enjoyed reading the added results and discussions of PTM analysis. Given the importance of PTM for phase phase separation, it will be meaningful to systematically explore how PTM impact the binding to G4. However, I do understand that this may be beyond the scope of this study. I look forward to seeing future study on this.

I added a few minor comments for clarity.

Line 158 and line 162: GO-term is first used in line 158 but defined in line 162. Please move it up and describe Gene Ontology.

Change the wording to emphasize the effort of developing G4-FUNNIES. An example is given below.

Line 227: We built a G4-Folded/UNfolded Nuclear Interaction Explorer System (G4-FUNNIES) employing the cleverMachine model as described above and made it publicly available at http://service.tartagliolab.com/new_submission/G4FUNNIES. Users can use it to to estimate the RNA G4-binding propensities of other proteins.

Please explain how statistical analysis is performed (i.e., AUC) in the CleverMachine method session.

Reviewer #1 (Remarks to the Author):

The authors have addressed all my concerns and substantially clarified the manuscript

We thank the Reviewer for the insightful comments

Reviewer #2 (Remarks to the Author):

The response fails to address the most significant concerns raised by me. Being central to the findings reported, lack of substantiation on these points limits the significance of the overall work.

#1. The number of RNA G-quadruplexes in the genome with stems of 4Gs instead of 3Gs is vastly different. I believe G-quads with 4Gs in the stem might be substantially less (possibly less than 1%) of G-quads with 3Gs in the stem in the human genome - this can be quickly checked to ascertain. Therefore, the scope of this study centred on stems comprising 4Gs is substantially limited in its usefulness. Authors address the point by showing 4G-stem Gquads are stable. That is hardly surprising. More number of Gs confer higher stability. Importantly, however, this was not the concern raised.

We agree with the Reviewer. The exact repetitions of GGG and GGGG sequences are infrequent in the human genome, as reported in databases such as QUADRAtlas. Consequently, we have incorporated a new paragraph in the introduction to address this point:

“We note that sequences such as G4A4 and G3A2²⁰, which form G-quadruplexes with four and three perfectly stacked quartets, are present in a minor proportion (2.5% and 25%, respectively) within the QUADRAtlas database¹³. Yet, G4A4 demonstrates greater stability compared to VEGFA¹⁹, and TERRA²⁸ G4, which is instrumental in determining its precise interactome.”

#2. Direct binding assays are imperative to give confidence to prediction studies/work. Binding studies performed, to address the point, show micromolar range of binding. Given the relatively weak binding, the possibility of non-specific binding should be ruled out using similar sequence(s) that cannot adopt the folded G-quad structure. This was not done, therefore, the results are again of limited significance.

We appreciate the suggestion and have accordingly conducted new pull-down experiments as direct binding assays to bolster the evidence for RUVBL2's affinity for G4 sequences. For these experiments, we utilized the G3A2 sequence, which includes GGG repetitions - previously recommended by the Reviewer - and the G3 mutant (G3 Mut) sequence as a control, which has been modified to prevent G4 structure formation. This approach is also employed in Herviou et al (<https://www.nature.com/articles/s41467-020-16168-x>), to demonstrate protein binding specificity to G4. We have decided to omit CHEK1 data to concentrate on a robust validation of a single candidate. This decision is elaborated in the Results section:

*“To validate whether the G4-FUNNIES candidates we assign as G4RBP bind preferentially to G4 RNA, we pulled down RUVBL2 from nuclear extracts using G4-forming G3A2 and its unstructured counterpart G3 Mut RNA (**Figure 6b; Online Methods**)²⁰. We observed a significantly higher enrichment of RUVBL2 when using the structured G4-forming G3A2 than with the G3 mutant that does not form structured G4 (**Figure 6c**), supporting that RUVBL2 binds to various G4-forming sequences and underlines the contribution of proper structure of the folded RNA for efficient binding to G4RBPs. Next, we assessed binding of RUVBL2 to G4 RNA sequences within the cell by native RNA immunoprecipitation followed by RT-qPCR analysis of protein-bound RNAs (**Figure 6d; Online Methods**).”*

Reviewer #3 (Remarks to the Author):

Authors addressed my comments from the first round of review. I particularly enjoyed reading the added results and discussions of PTM analysis. Given the importance of PTM for phase phase separation, it will be meaningful to systematically explore how PTM impact the binding to G4. However, I do understand that this may be beyond the scope of this study. I look forward to seeing future study on this.

We thank the Reviewer for the insightful comments on the PTM analysis. The suggestion to systematically explore PTM's impact on G4 binding is indeed intriguing and important for understanding phase separation. While it may extend beyond the current study's scope, it certainly outlines a promising direction for future research. We are encouraged by your interest and look forward to potentially delving into this area in subsequent work.

I added a few minor comments for clarity.

Line 158 and line 162: GO-term is first used in line 158 but defined in line 162. Please move it up and describe Gene Ontology.

We correct this accordingly.

Change the wording to emphasize the effort of developing G4-FUNNIES. An example is given below.

Line 227: We built a G4-Folded/UNfolded Nuclear Interaction Explorer System (G4-FUNNIES) employing the cleverMachine model as described above and made it publicly available at [We are grateful for this comment. We have changed the text into:](https://urldefense.com/v3/_http://service.tartaglialab.com/new_submission/G4FUNNIES_!!NLFgqXoFfo8MMQ!qHzXFQYwU15qeCPUEKnfRwGX7JfYXFvQSPuO46dNjp1EvnZKbKRCvo9bZFa7Zrm6wvRoNdGOWFvqm1DiZCtINjn9e-hfJA$. Users can use it to to estimate the RNA G4-binding propensities of other proteins.<div data-bbox=)

"We developed the G4-Folded/UNfolded Nuclear Interaction Explorer System (G4-FUNNIES) using the cleverMachine model and have made this resource publicly available. The users can access G4-FUNNIES to evaluate RNA G4-binding propensities of proteins via the following link: http://service.tartaglialab.com/new_submission/G4FUNNIES."

Please explain how statistical analysis is performed (i.e., AUC) in the CleverMachine method session.

Accordingly, we have added a paragraph that complements the information already available in the cleverMACHINE webpages:

"In the cleverMACHINE classification, the three-scale combination (classical⁷⁹ and nonclassical⁵ RNA-binding ability as well as burial energy⁸⁰) achieved a True Positive Rate (TPR) of 0.99, False Positive Rate (FPR) of 0.06, and an MCC of 0.907, with the highest cross-validation accuracy of 0.96. The five-scale combination (including hydrophobicity⁸¹ and aggregation⁸²) showed a TPR of 1.00, FPR of 0.05, and an MCC of 0.928, but a slightly lower cross-validation accuracy of 0.91. This indicates the three-scale model's superior predictive capability in training for distinguishing protein interactions under K⁺ and Li⁺ conditions. Further details on the statistics related to the cleverMACHINE approach are available at http://service.tartaglialab.com/static_files/algorithms/clever_machine/documentation.html and http://service.tartaglialab.com/static_files/algorithms/clever_machine/tutorial.html."